# Phrenic-specific transcriptional programs shape respiratory motor output

Alicia N Vagnozzi[1], Kiran Garg[1], Carola Dewitz[2], Matthew T Moore[1], Jared M Cregg[1], Lucie Jeannotte[3], Niccolò Zampieri[2], Lynn T Landmesser[1], Polyxeni Philippidou[1]*

[1]Department of Neurosciences, Case Western Reserve University School of Medicine, Cleveland, United States; [2]Max Delbrück Center for Molecular Medicine in the Helmholtz Association (MDC), Berlin, Germany; [3]Centre de Recherche sur le Cancer de l'Université Laval, Centre de recherche du CHU de Québec-Université Laval (Oncology), Québec, Canada

**Abstract** The precise pattern of motor neuron (MN) activation is essential for the execution of motor actions; however, the molecular mechanisms that give rise to specific patterns of MN activity are largely unknown. Phrenic MNs integrate multiple inputs to mediate inspiratory activity during breathing and are constrained to fire in a pattern that drives efficient diaphragm contraction. We show that Hox5 transcription factors shape phrenic MN output by connecting phrenic MNs to inhibitory premotor neurons. *Hox5* genes establish phrenic MN organization and dendritic topography through the regulation of phrenic-specific cell adhesion programs. In the absence of *Hox5* genes, phrenic MN firing becomes asynchronous and erratic due to loss of phrenic MN inhibition. Strikingly, mice lacking *Hox5* genes in MNs exhibit abnormal respiratory behavior throughout their lifetime. Our findings support a model where MN-intrinsic transcriptional programs shape the pattern of motor output by orchestrating distinct aspects of MN connectivity.

**\*For correspondence:**
pxp282@case.edu

**Competing interests:** The authors declare that no competing interests exist.

## Introduction

Breathing is a fundamental motor behavior required for life. In mammals, specialized circuits have evolved to support robust and efficient breathing to cope with changing metabolic demands. While respiratory rhythmogenesis occurs within a small neuronal kernel in the pre-Bötzinger complex, a large contingent of respiratory circuitry is dedicated to transforming this rhythm into precisely patterned motor output for efficient muscle contraction (*Feldman et al., 2013*). Phrenic Motor Column (PMC) neurons are a critical node in these circuits, as they integrate multiple descending and local inputs to mediate rhythmic contraction of the diaphragm muscle, which is essential for driving airflow into the lungs during inspiration (*Greer, 2012*). The establishment of phrenic motor neuron (MN) identity relies on the intersection of transcription factor-based programs acting along the dorsoventral and rostrocaudal axes of the spinal cord during development (*Chaimowicz et al., 2019*; *Edmond et al., 2017*; *Philippidou et al., 2012*). However, it is not known whether these MN-intrinsic transcriptional programs are required for the generation of patterned respiratory motor output.

While most neural circuits undergo significant maturation at postnatal stages, the circuits that underlie breathing not only need to be functional at birth, but are also critically required in utero. Fetal contractions of the diaphragm muscle are essential for lung and diaphragm development and defects in these contractions result in lung hypoplasia and central sleep apneas (*Harding, 1997*). Accordingly, inputs that drive the activation of phrenic MNs must be established prior to birth, indicating that hardwired transcriptional programs likely control major aspects of this connectivity. Because inspiratory drive is relatively weak during embryonic stages, coupling of phrenic MNs through gap junctions is critical to maintain early diaphragm contractions (*Greer and Funk, 2005*).

**eLife digest** In mammals, air is moved in and out of the lungs by a sheet of muscle called the diaphragm. When this muscle contracts air gets drawn into the lungs and as the muscle relaxes this pushes air back out. Movement of the diaphragm is controlled by a group of nerve cells called motor neurons which are part of the phrenic motor column (or PMC for short) that sits within the spinal cord. The neurons within this column work together with nerve cells in the brain to coordinate the speed and duration of each breath.

For the lungs to develop normally, the neurons that control how the diaphragm contracts need to start working before birth. During development, motor neurons in the PMC cluster together and connect with other nerve cells involved in breathing. But, despite their essential role, it is not yet clear how neurons in the PMC develop and join up with other nerve cells.

Now, Vagnozzi et al. show that a set of genes which make the transcription factor Hox5 control the position and organization of motor neurons in the PMC. Transcription factors work as genetic switches, turning sets of genes on and off. Vagnozzi et al. showed that removing the Hox5 transcription factors from motor neurons in the PMC changed their activity and disordered their connections with other breathing-related nerve cells. Hox5 transcription factors regulate the production of proteins called cadherins which join together neighboring cells. Therefore, motor neurons lacking Hox5 were unable to make enough cadherins to securely stick together and connect with other nerve cells.

Further experiments showed that removing the genes that code for Hox5 caused mice to have breathing difficulties in the first two weeks after birth. Although half of these mutant mice were eventually able to breathe normally, the other half died within a week. These breathing defects are reminiscent of the symptoms observed in sudden infant death syndrome (also known as SIDS).

Abnormalities in breathing occur in many other diseases, including sleep apnea, muscular dystrophy and amyotrophic lateral sclerosis (ALS). A better understanding of how the connections between nerve cells involved in breathing are formed, and the role of Hox5 and cadherins, could lead to improved treatment options for these diseases.

Therefore, tight clustering of PMC neurons is essential for proper development of the respiratory system, but the molecular mechanisms that control phrenic MN clustering and topography are largely unknown.

In addition to establishing robust embryonic diaphragm contractions, PMC clustering may also be important for the selective establishment of presynaptic inputs. Interestingly, unlike the majority of MNs in the spinal cord, phrenic MNs appear to lack monosynaptic sensory input from Ia proprioceptive afferents and extensive inputs from spinal cord interneurons (*Wu et al., 2017*). Instead, the majority of their monosynaptic inputs (~70%) are derived from the rostral Ventral Respiratory Group (rVRG), a group of mostly glutamatergic neurons in the brainstem that transmit the excitatory drive to activate MNs in a rhythmic pattern generated in the pre-Bötzinger complex (*Wu et al., 2017*). Inhibition also contributes to shaping phrenic motor output. Inspiratory-phase inhibitory inputs onto phrenic MNs rapidly modulate MN excitability to act as a gain control for motor output and control motor neuron synchronization within respiratory bursts (*Parkis et al., 1999*). Synchronous inspiratory MN firing on a short time scale is thought to enhance inspiratory muscle activation, thus ensuring robust and efficient breathing (*Bou-Flores and Berger, 2001*). Importantly, firing of phrenic MNs in an unpatterned manner increases the risk of phrenic MN adaptation, diaphragm muscle fatigue, and respiratory failure (*Martin-Caraballo et al., 2000*). Despite the critical role of inhibition in patterning phrenic MN activity, the molecular determinants that underlie the establishment of inhibitory inputs onto phrenic MNs have not been identified.

Here, we show that Hox5 transcription factors are required for robust and efficient breathing. *Hox5* loss renders mice particularly vulnerable to respiratory dysfunction in the first two weeks of life, suggesting that *Hox5* mutations may contribute to early life respiratory conditions. We also show that Hox5 proteins establish phrenic MN clustering and topography through the regulation of a network of cell adhesion molecules. We find that a subset of cadherins are specifically expressed in phrenic MNs and that loss of cadherin function through conditional disruption of downstream β/γ-

catenin signaling leads to phrenic MN cell body disorganization and dendrite displacement. MN-specific deletion of *Hox5* genes results in a selective loss of inhibitory inputs to PMC neurons and a dramatic change in the activation pattern of phrenic MNs. Our results demonstrate that Hox5 transcription factors determine phrenic MN topography and connectivity to generate robust breathing behaviors.

## Results

### *Hox5* genes are required for proper respiratory behavior

We previously showed that two *Hox5* paralogs, *Hoxa5* and *Hoxc5* (collectively referred to as *Hox5* genes) are required for the early development and survival of phrenic MNs and the innervation of the diaphragm (*Landry-Truchon et al., 2017*; *Philippidou et al., 2012*). Mice lacking both *Hox5* genes in MNs (*Hoxa5 flox/flox; Hoxc5-/-; Olig2::Cre*, referred to as *Hox5^{MNΔ}* mice) die at birth due to respiratory defects (*Philippidou et al., 2012*). While the neonatal lethality of *Hox5^{MNΔ}* mice underscores the critical requirement for *Hox5* genes in respiration, it had prevented us from examining the role of Hox5 proteins in respiratory behaviors and functional connectivity at postnatal stages. To examine the role of *Hox5* genes in breathing behaviors and phrenic MN output over time we utilized an alternative mouse model. We generated mice in which a single *Hox5* gene, *Hoxa5* was selectively deleted from MNs by crossing a conditional *Hoxa5* allele (*Tabariès et al., 2007*) to *Olig2::Cre* mice (*Hoxa5^{MNΔ}*) (*Dessaud et al., 2007*). *Hoxa5^{MNΔ}* mice were viable, and we therefore introduced the *Hoxa5^{MNΔ}* mutant allele into a *Hoxc5* heterozygous background. Mice lacking *Hoxa5* specifically from MNs and a single copy of *Hoxc5* (*Hoxa5 flox/flox; Hoxc5+/-; Olig2::Cre*, referred to as *Hoxa5^{MNΔ}; c5^{het}* mice) exhibit a 60% reduction in total diaphragm motor innervation (*Figure 1—figure supplement 1a–b*). Importantly, around 50% of *Hoxa5^{MNΔ}; c5^{het}* mice survive to adulthood, enabling us to examine how MN-specific *Hox5* loss impacts respiration and phrenic MN output.

In order to assess breathing in *Hoxa5^{MNΔ}; c5^{het}* mice, we utilized unrestrained whole body flow-through plethysmography (*Figure 1a*). We found that adult (P80) *Hoxa5^{MNΔ}; c5^{het}* mice show a 40% decrease in tidal volume (the amount of air inhaled during a normal breath), accompanied by a compensatory increase in respiratory frequency (*Figure 1b–c*, *Figure 1—figure supplement 2a–b*). The increased frequency allows *Hoxa5^{MNΔ}; c5^{het}* mice to breathe in an equal volume of air per minute (minute ventilation) as control animals (*Figure 1c*, *Figure 1—figure supplement 2c*), albeit at the cost of expending more energy. We next submitted mice to a hypercapnic challenge (5% $CO_2$, *Figure 1d*). In control (*Hoxa5 flox/flox; Hoxc5+/-*) mice, exposure to 5% $CO_2$ results in an increase in tidal volume and breathing frequency. *Hoxa5^{MNΔ}; c5^{het}* mice only slightly increase their tidal volume during hypercapnia, resulting in a striking 30% decrease in ventilation and a diminished capacity to respond to respiratory challenges (*Figure 1e–f*, *Figure 1—figure supplement 2d–f*).

Notably, the ability to overcome hypercapnic and hypoxic conditions is particularly important during perinatal stages, as a compromised response to respiratory insults could increase susceptibility to sudden infant death syndrome (SIDS) (*Kinney et al., 2009*). We recorded breathing in unrestrained control and *Hoxa5^{MNΔ}; c5^{het}* mice at different ages to temporally characterize respiratory dysfunction in the absence of *Hox5* genes. We found that tidal volume is consistently decreased in *Hoxa5^{MNΔ}; c5^{het}* pups as early as P3 (*Figure 1g*); however, the compensatory increase in breathing frequency does not develop until approximately 2 weeks after birth (*Figure 1h*). Thus, perinatal *Hoxa5^{MNΔ}; c5^{het}* mice exhibit a severe (50–60%) reduction in ventilation, even at rest in normal air, for the first two weeks of life (*Figure 1i*). Consistent with this, we find that approximately 50% of neonatal *Hoxa5^{MNΔ}; c5^{het}* mice perish within a week after birth. Altogether, our plethysmography results demonstrate that *Hox5* genes are required for the emergence of proper respiratory behavior. Importantly, we have identified a critical temporal window during which *Hoxa5^{MNΔ}; c5^{het}* mice are especially vulnerable to respiratory dysfunction, similar to perinatal susceptibility to SIDS.

### *Hox5* genes dictate phrenic MN clustering and dendritic topography

What are the molecular underpinnings of respiratory dysfunction in *Hoxa5^{MNΔ}; c5^{het}* mice? While the reduction in diaphragm innervation likely contributes to tidal volume changes, we wanted to identify additional fundamental properties of phrenic MNs that could contribute to respiratory circuit formation and function that are altered in *Hoxa5^{MNΔ}; c5^{het}* mice. Phrenic MNs form a tightly-packed

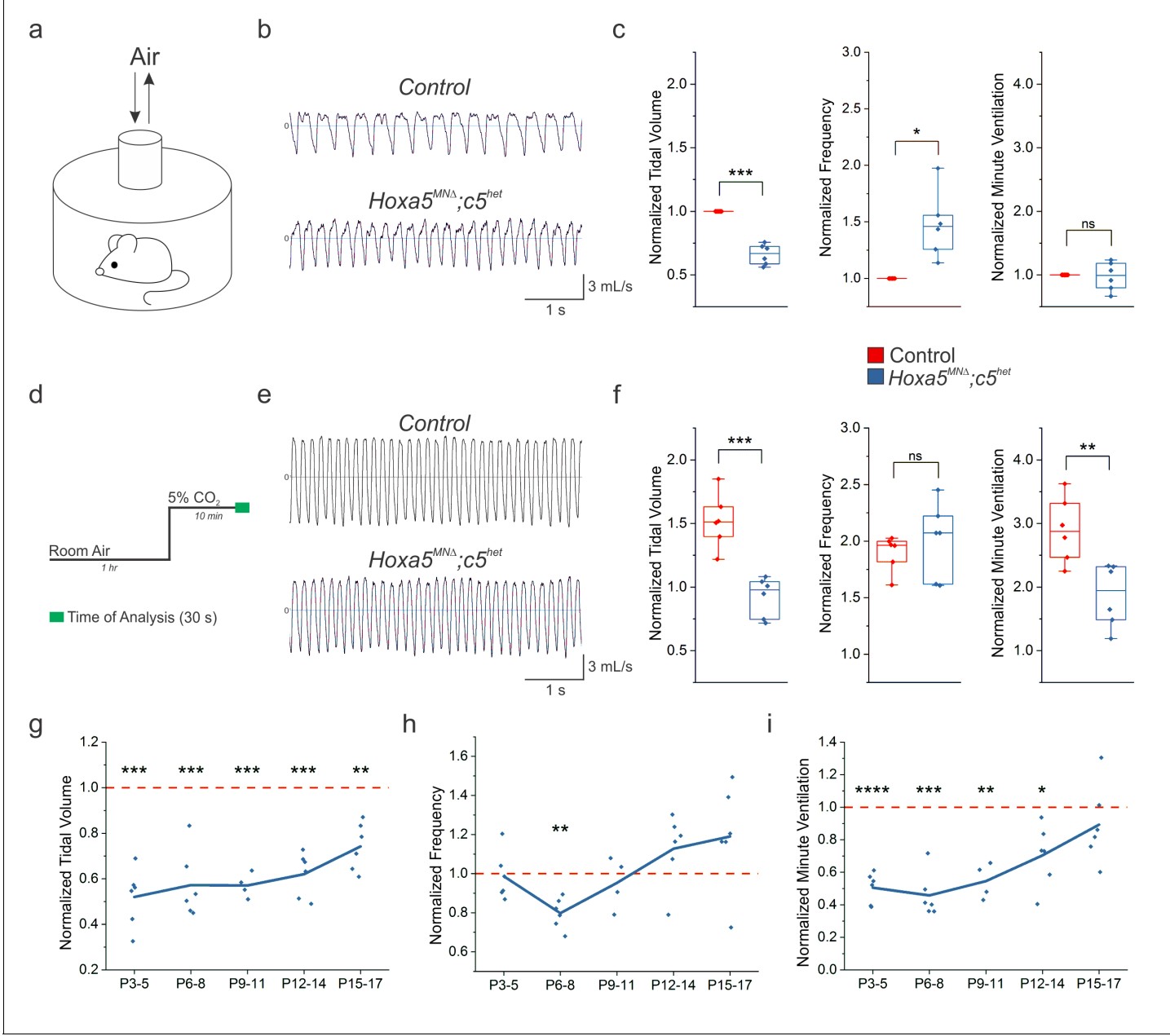

**Figure 1.** *Hox5* genes are required for proper respiratory behavior throughout life. (a) Experimental setup for unrestrained whole body plethysmography experiments. (b) Example respiratory traces in room air from adult (P80) control and *Hoxa5^MNΔ; c5^het* mice. (c) *Hoxa5^MNΔ; c5^het* mice breathe shallow (decreased tidal volume), fast (increased frequency) breaths in room air, but maintain overall ventilation (n = 6 for each genotype). (d) Time course for 5% $CO_2$ exposure experiments. (e) Example respiratory traces in 5% $CO_2$ from adult (P80) control and *Hoxa5^MNΔ; c5^het* mice. (f) Control mice increase the depth and frequency of breathing in response to 5% $CO_2$, but in *Hoxa5^MNΔ; c5^het* mice the tidal volume increase is blunted and overall ventilation is decreased. (g–i) Temporal analysis of neonatal respiration. Tidal volume in room air is decreased at all neonatal ages (P3–P17) in *Hoxa5^MNΔ; c5^het* mice, but the compensatory increase in frequency does not develop until approximately P16. Therefore, *Hoxa5^MNΔ; c5^het* mice have severe respiratory insufficiency even in room air for the first two weeks of life (n = 4 for each genotype at P9-11, all other timepoints n = 6). See also *Figure 1—figure supplements 1* and *2*.

The online version of this article includes the following figure supplement(s) for figure 1:

**Figure supplement 1.** *Hox5* genes control diaphragm innervation.
**Figure supplement 2.** Respiratory insufficiency in both female and male *Hoxa5^MNΔ; c5^het* mice.

neuronal cluster at cervical levels of the spinal cord (C3-C5). This clustering is critical for the proper development of the respiratory system because it facilitates recruitment of motor units through electrical coupling in the embryo to compensate for weak inspiratory drive (*Greer and Funk, 2005*). In addition, the stereotypical orientation of phrenic dendrites likely enables their targeting by premotor respiratory neurons. However, the mechanisms that control PMC clustering and dendritic topography have not been established.

To examine whether phrenic MN organization and dendritic orientation are altered in *Hoxa5^MNΔ^; c5^het^* mice, we injected the lipophilic dye diI into phrenic nerves at e18.5. diI diffuses along the phrenic nerve to label both PMC cell bodies in the spinal cord and axons innervating the diaphragm. To ensure that we labeled the full extent of the PMC we only analyzed mice in which all diaphragm projections were labeled (*Figure 2—figure supplement 1a–b*). In control mice, retrogradely-labelled PMC neurons are found in close proximity to each other, with no distance in between, however in *Hoxa5^MNΔ^; c5^het^* mice phrenic MNs lose their stereotypical clustering organization (*Figure 2a*, *Figure 2—figure supplement 1c–d*). To quantitate the impact of *Hox5* loss on PMC organization, we established a clustering index, representing the percentage of PMC neurons in contact with each other (1 = 100%, see Materials and methods). We found a significant reduction in the clustering index of *Hoxa5^MNΔ^; c5^het^* mice, indicating that *Hox5* genes control phrenic MN clustering (*Figure 2b*).

To verify that the MN disorganization observed upon *Hox5* deletion was not due to the progressive loss of PMC neurons by e18.5 (*Figure 2—figure supplement 1e*), we assessed PMC clustering at earlier developmental time points. We observed a similar disorganization of phrenic MNs, identified by the expression of the MN-specific transcription factor Isl1/2 and the phrenic-specific transcription factor Scip, in *Hoxa5^MNΔ^; c5^het^* mice at e12.5 (*Figure 2—figure supplement 2a–d*). We also introduced the *Hox5^MNΔ^* allele (*Hoxa5 flox/flox; Hoxc5-/-; Olig2::Cre*) into a *Bax* mutant background to prevent programmed cell death (*Knudson et al., 1995*; *Philippidou et al., 2012*). Since *Bax* deletion circumvents phrenic MN loss in the absence of *Hox5* genes, we performed this analysis in *Hox5^MNΔ^* mice to define the impact of complete *Hox5* loss of function on PMC organization. Deletion of *Bax*, both in control and in *Hox5^MNΔ^* (*Hox5^MNΔ^; Bax-/-*) mice, dramatically increased the number of PMC neurons. However, phrenic MNs still appeared to lose their tight clustering in *Hox5^MNΔ^; Bax-/-* mice, demonstrating that *Hox5* genes drive a program that controls PMC organization and clustering independently of phrenic MN survival (*Figure 2—figure supplement 2e–h*).

In addition to the loss of phrenic MN clustering, we also observed a remarkable redistribution of phrenic MN dendrites in *Hoxa5^MNΔ^; c5^het^* mice after diI injection. While in control mice phrenic dendrites project in two major directions, dorsolateral and ventromedial, in *Hoxa5^MNΔ^; c5^het^* mice the most dorsal dendritic projections are lost and dendrites appear defasciculated (*Figure 2c*). To quantitate the change in dendritic orientation, we established a grid separating the area proximal to phrenic cell bodies in eight squares and measured the percentage of labelled dendrites in each square (*Figure 2c–e*, *Figure 2—figure supplement 1g–h*, see Materials and methods). We found a significant decrease in dorsolateral dendrites in *Hoxa5^MNΔ^; c5^het^* mice, as well as an increase in the number of dendrites approaching the midline (*Figure 2c–e*). While in control mice phrenic dendrites rarely cross the midline, in *Hoxa5^MNΔ^; c5^het^* mice we find that a number of dendrites traverse the midline and continue to grow contralaterally, despite dendrites covering less area overall (*Figure 2f–h*, *Figure 2—figure supplement 1f*). Our data indicate that *Hox5* genes control phrenic MN dendritic topography, which likely contributes to their presynaptic targeting.

## A Hox5-dependent PMC-specific cell adhesion molecular code

In order to understand how Hox5 transcription factors regulate phrenic MN topography and clustering, we analyzed the transcriptome after *Hox5* loss using RNA sequencing (RNA-seq), taking advantage of *Hb9::GFP* mice, which selectively express GFP in MNs (*Wichterle et al., 2002*). Using fluorescence activated cell sorting (FACS), we isolated RNA from GFP-positive MNs from the cervical spinal cord (C3-C6, which encompasses the entire PMC) of control (*Bax-/-*) and *Hox5^MNΔ^; Bax-/-* mice at e12.5 (*Figure 3a*). We performed RNA-seq on isolated GFP+ MNs in *Bax-/-* and *Hox5^MNΔ^; Bax-/-* mice to identify gene expression changes independently of phrenic MN cell death that begins at e12.5. Since *Bax* deletion circumvents phrenic MN loss in the absence of *Hox5* genes, we performed this analysis in *Hox5^MNΔ^* mice to maximize gene expression changes. Our analysis identified 837 genes that were differentially expressed between *Bax-/-* and *Hox5^MNΔ^; Bax-/-* mice (p<0.05,

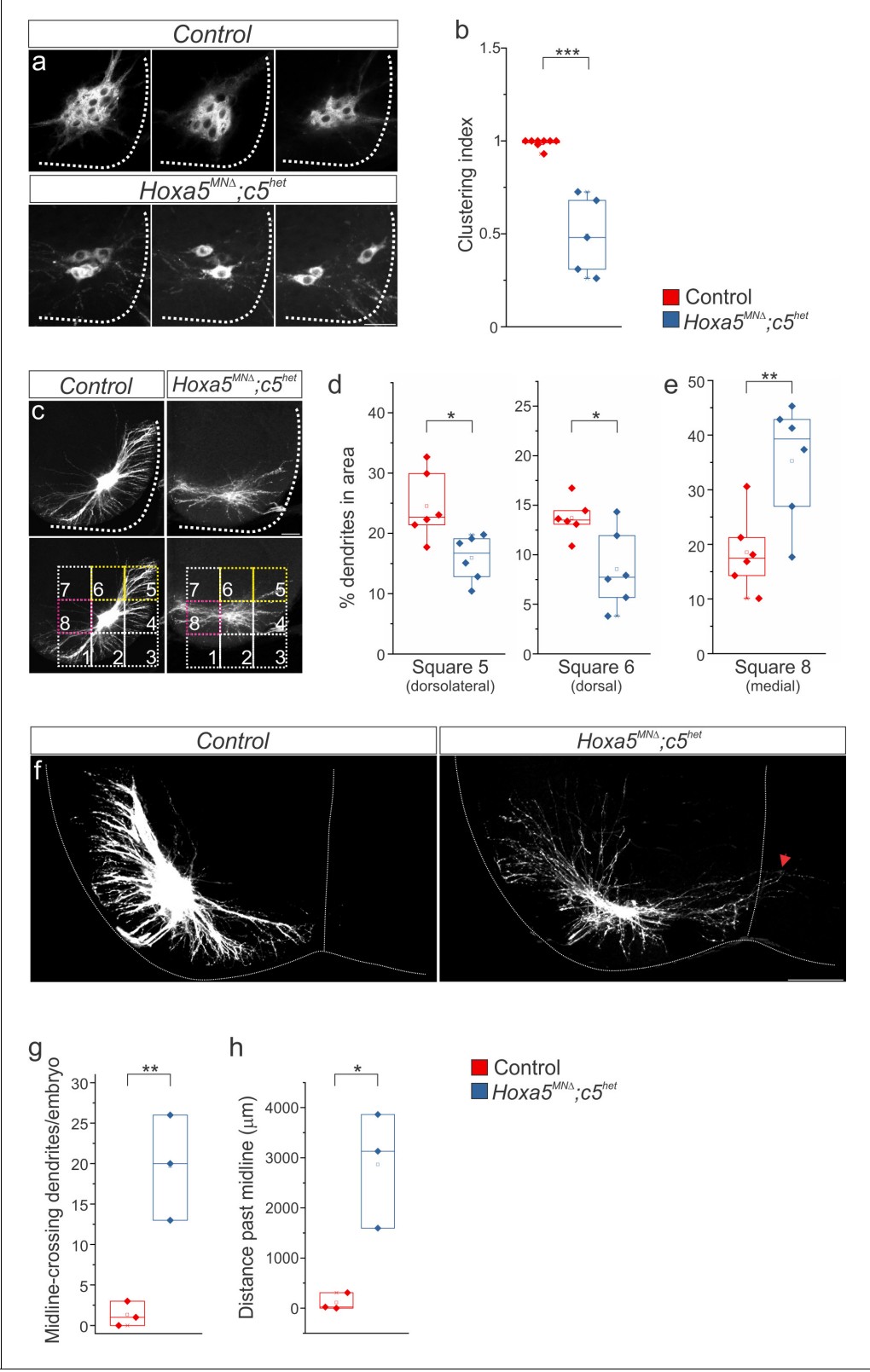

**Figure 2.** *Hox5* genes dictate phrenic MN organization and dendritic topography. diI injections into the phrenic nerve show loss of PMC clustering (**a–b**) and changes in dendritic orientation (**c–e**) in *Hoxa5*^MNΔ^; *c5*^het^ mice at e18.5. (**c**) A numbered grid was superimposed on PMC neurons to measure changes in dendritic orientation (see also Materials and methods). (**d–e**) Quantification of dendritic orientation. (**d**) Loss of dorsolateral dendrites (squares 5+6, yellow in c) in *Hoxa5*^MNΔ^; *c5*^het^ mice. (**e**) Increase in the percentage of dendrites projecting towards the midline (square 8, pink in c) in

*Figure 2 continued on next page*

*Hoxa5^{MNΔ}; c5^{het}* mice. (**f–h**) Phrenic dendrites in *Hoxa5^{MNΔ}; c5^{het}* mice cross the midline more frequently (**g**) and for longer distances (**h**) than in control mice. Scale bar = 50 μm in a, 100 μm in c, f. See also *Figure 2—figure supplements 1* and *2*.

The online version of this article includes the following figure supplement(s) for figure 2:

**Figure supplement 1.** *Hox5* genes dictate phrenic MN organization.
**Figure supplement 2.** *Hox5* genes control PMC position and clustering.

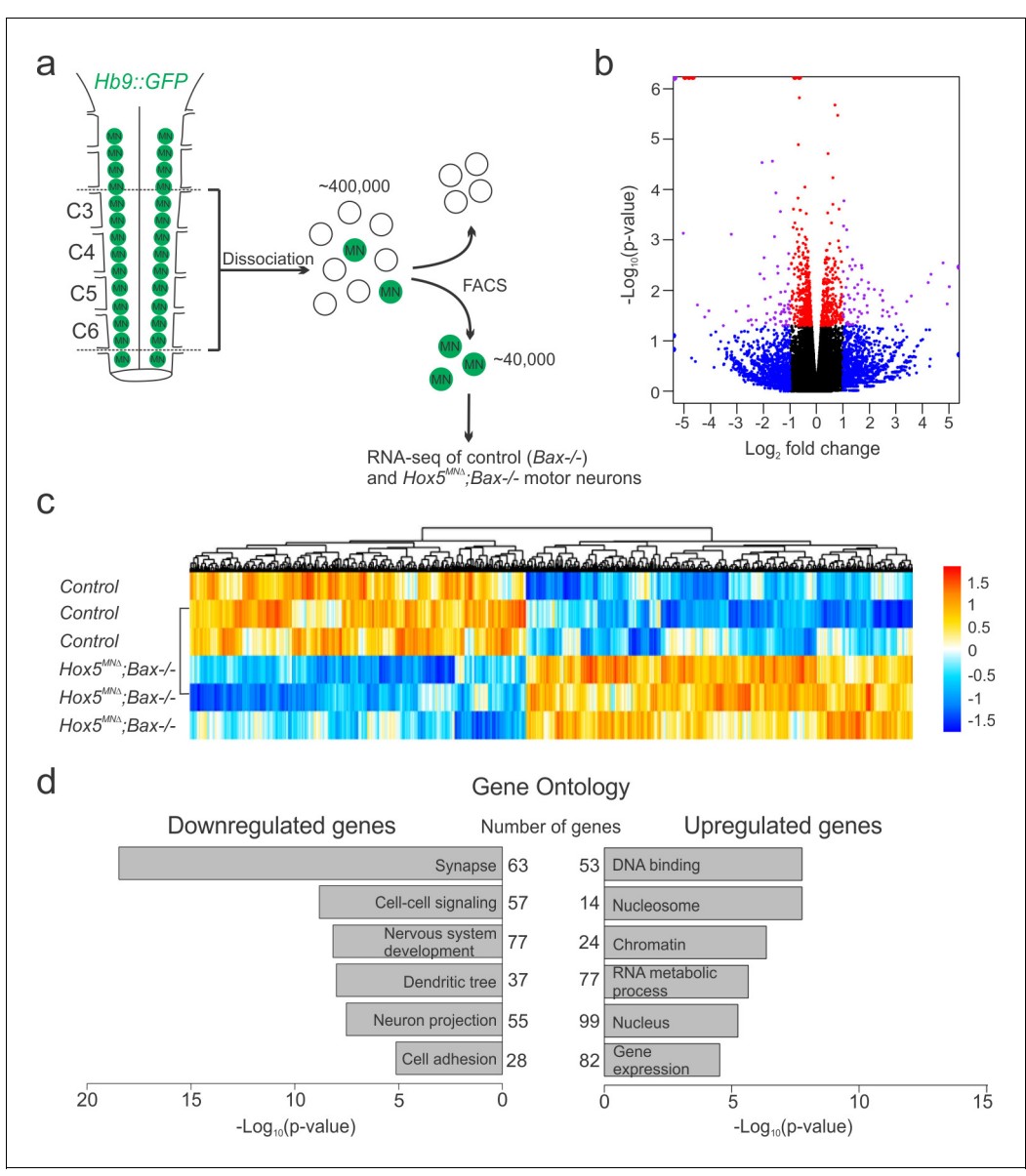

**Figure 3.** Hox5-regulated MN transcriptome. (**a**) Experimental design for RNA-seq experiments. MNs were FAC-sorted from control and *Hox5^{MNΔ}; Bax-/-* mice in an *Hb9::GFP* background at e12.5. (**b–c**) Volcano plot (**b**) and heat map (**c**) showing differential gene expression between control and *Hox5^{MNΔ}; Bax-/-* mice. (**d**) Gene ontology analysis reveals a number of genes involved in nervous system development, including cell adhesion molecules, are downregulated in *Hox5^{MNΔ}; Bax-/-* mice. See also *Figure 3—figure supplement 1*.

The online version of this article includes the following figure supplement(s) for figure 3:

**Figure supplement 1.** Hox5-regulated cell adhesion molecules.

*Figure 3b–c*). Gene ontology analysis revealed that the majority of downregulated genes were implicated in processes relevant to nervous system development, including neuron projections and dendritic and synapse development (*Figure 3d*).

Since *Hox5* genes control PMC clustering and dendritic orientation, we reasoned that cell adhesion molecules (CAMs) might be good candidate Hox5 downstream effectors. Therefore, we performed in situ hybridization to validate whether CAMs identified in our RNA-seq (*Figure 3—figure supplement 1*) showed Hox5-dependent PMC-specific expression. We found that *ALCAM*, *Edil3*, *cdh9*, *Ptprt*, *Lsamp* and *Negr1* were highly and specifically expressed in the PMC at e12.5 (*Figure 4a*). A subset of these CAMs were previously established as phrenic-specific markers (*ALCAM*, *Edil3*, *cdh9* and *Negr1*), while our analysis also identified novel PMC genes (*Ptprt* and *Lsamp*) (*Machado et al., 2014*). We further found that these phrenic-specific CAMs require Hox5 proteins for their expression, as they were downregulated in $Hox5^{MN\Delta}$; *Bax-/-* mice (*Figure 4a*). This downregulation was further recapitulated in both $Hoxa5^{MN\Delta}$; $c5^{het}$ mice and $Hoxa5^{MN\Delta}$; $c5^{het}$; *Bax-/-* mice (*Figure 4—figure supplement 1a*), indicating that a single copy of *Hoxc5* is insufficient to induce PMC-specific CAM expression. Our results suggest that Hox5 proteins regulate PMC clustering and position through controlling the expression of a network of downstream cell adhesion molecules.

Based on our RNA-seq analysis and validation, we identified *cdh9* as the PMC-specific CAM that was most downregulated after *Hox5* deletion (*Figure 3—figure supplement 1*). In the spinal cord, cadherin function is required for the segregation and clustering of limb-innervating MNs into nuclear structures called pools, however the role of cadherins in respiratory motor neurons has not been examined (*Price et al., 2002*). We asked whether cadherins might play a role in PMC organization. First, we wanted to define the full repertoire of cadherin expression in the PMC. We performed in situ hybridization for all type I and type II cadherins and found that *cdh2, 6, 9, 10, 11* and *22* were expressed in the PMC at e13.5, while the rest of the family members were either expressed in other MN subtypes or not found in the spinal cord (*Figure 4—figure supplement 1b–c*). To further quantify cadherin expression in phrenic MNs, we performed fluorescence in situ hybridization for PMC-enriched cadherins combined with immunofluorescence for the phrenic marker Scip. We found that all PMC cadherins were uniformly expressed in the majority of phrenic MNs (>90%) at e13.5 (*Figure 4b–c*, *Figure 4—figure supplement 1d*). At cervical levels of the spinal cord, cdh9 and 10 expression appears to be restricted to phrenic MNs, while cdh2, 6, 11 and 22 are broadly expressed in all MN populations (*Figure 4b–c*). Our data establish a comprehensive combinatorial cadherin code that uniquely defines PMC neurons.

## Cadherins establish PMC organization and dendritic orientation

The highly specific PMC cadherin expression pattern suggests that cadherins could have important functions in phrenic MNs. In order to assess the role of classical cadherins in PMC development, we eliminated their function by inactivating β- and γ-catenin specifically in MNs using an *Olig2::cre* promoter (*β-catenin flox/flox;γ-catenin flox/-;Olig2::cre*, referred to as $\beta\gamma$-$cat^{MN\Delta}$ mice) (*Figure 5a*). β- and γ-catenin are obligate intracellular factors required for cadherin-mediated cell adhesive function and are necessary for the organization of limb-innervating motor pools (*Demireva et al., 2011*). The strategy of inactivating β/γ-catenin in MNs circumvents potential redundancy that can arise through the expression of multiple cadherins in the PMC and allows us to establish a cadherin requirement in PMC development before dissecting individual cadherin function.

Single β- or γ-catenin mutants exhibited normal phrenic MN numbers, cell body position and clustering (*Figure 5—figure supplement 1b–c*), indicating that disruption of Wnt signaling through *β-catenin* inactivation does not affect PMC topography. Joint inactivation of β- and γ-catenin, however, and disruption of cadherin signaling, led to a marked disorganization and loss of phrenic MN clustering (*Figure 5b–c*). We find that the PMC clustering index is significantly reduced in $\beta\gamma$-$cat^{MN\Delta}$ mice at e13.5 (*Figure 5c*). We also observed a ~30% reduction in the number of Scip+ MNs that settle in the ventral spinal cord in $\beta\gamma$-$cat^{MN\Delta}$ mice (*Figure 5—figure supplement 1b*), partly due to the failure of a subset of PMC neurons to migrate away from the midline (*Figure 5—figure supplement 1d*).

Next, we examined whether cadherins also play a role in PMC dendritic orientation. Since $\beta\gamma$-$cat^{MN\Delta}$ mice die around e14.5-e15.5 (*Demireva et al., 2011*), we performed diI phrenic nerve injections at e14.5. Similar to $Hoxa5^{MN\Delta}$; $c5^{het}$ mice, we found a change in dendritic orientation and loss

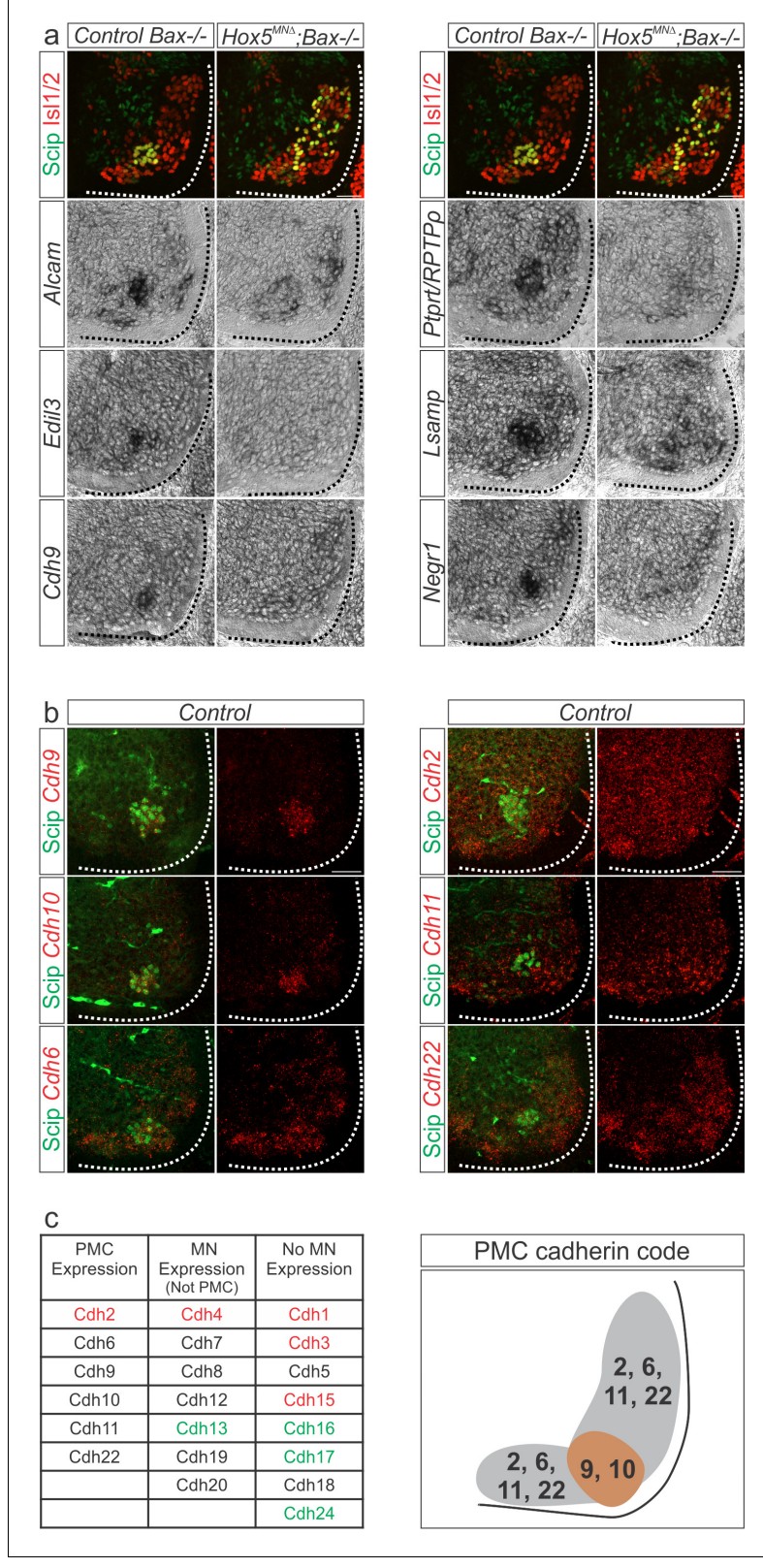

**Figure 4.** A Hox5-dependent PMC-specific cell adhesion molecule code. (a) Validation of Hox5-regulated cell adhesion molecules by in situ hybridization at e12.5. (b) Cadherin expression in the PMC at e13.5. Fluorescence in situ hybridization for the indicated cadherin (red) was combined with antibody staining for the phrenic-specific transcription factor Scip (green). (c) Comprehensive analysis of cadherin expression in the cervical/brachial spinal

*Figure 4 continued on next page*

*Figure 4 continued*

cord at e13.5. Type I cadherins are indicated in red, type II in black and atypical cadherins in green. PMC neurons can be defined by the combinatorial expression of cadherin 9 and 10 that are restricted to the PMC at cervical levels and cadherin 2, 6, 11 and 22 that are broadly expressed in MNs. Scale bar = 50 µm. See also *Figure 4—figure supplement 1*.

The online version of this article includes the following figure supplement(s) for figure 4:

**Figure supplement 1.** A Hox5-dependent PMC-specific cell adhesion molecule code.

of the dorsal-most dendrites in $\beta\gamma\text{-}cat^{MN\Delta}$ mice (*Figure 5d–e*, *Figure 5—figure supplement 1e*). Unlike in *Hoxa5*$^{MN\Delta}$; $c5^{het}$ mice however, phrenic dendrites do not cross the midline in $\beta\gamma\text{-}cat^{MN\Delta}$ mice, suggesting that multiple pathways are acting downstream of Hox5 proteins to dictate precise phrenic dendritic orientation. Despite having a striking effect on PMC dendrites, joint inactivation of $\beta$- and $\gamma$-catenin did not affect phrenic axon growth or guidance, as diaphragm innervation appears normal (*Figure 5—figure supplement 1a*), indicating that *Hox5* genes control phrenic axon and dendrite development through distinct molecular programs. Our results demonstrate that cadherins

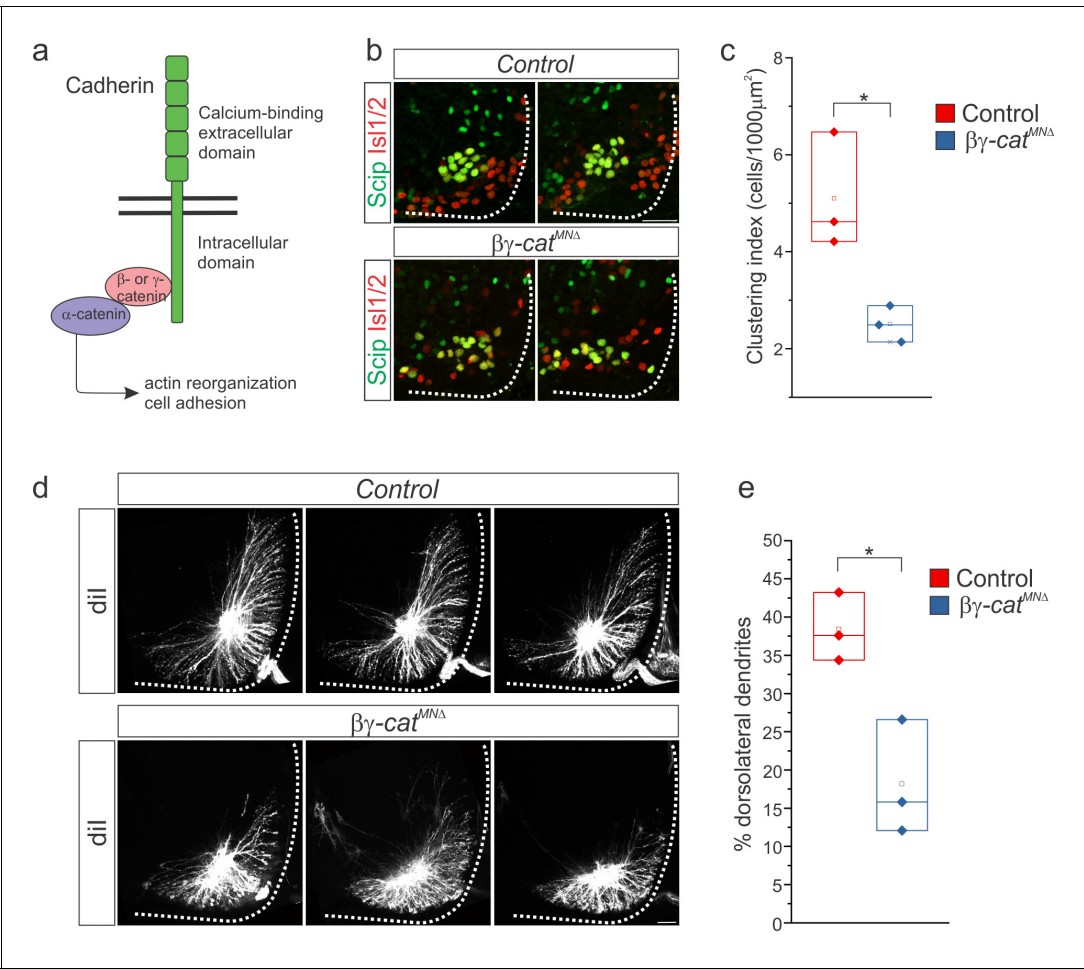

**Figure 5.** $\beta$- and $\gamma$-catenin inactivation leads to phrenic cell body and dendrite disorganization. (a) Inactivation of $\beta$- and $\gamma$-catenin in MNs as a strategy to investigate the role of cadherins in PMC organization. (b–c) PMC disorganization and loss of clustering in $\beta\gamma\text{-}cat^{MN\Delta}$ mice at e13.5. (d–e) Loss of dorsolateral dendrites in $\beta\gamma\text{-}cat^{MN\Delta}$ mice at e14.5. Images in b and d show the PMC of the same embryo at different rostrocaudal levels, in a rostral to caudal order. Scale bar = 50 µm. See also *Figure 5—figure supplement 1*.

The online version of this article includes the following figure supplement(s) for figure 5:

**Figure supplement 1.** $\beta$- and $\gamma$-catenin inactivation leads to phrenic cell body and dendrite disorganization.

are required in phrenic MNs downstream of *Hox5* genes for proper clustering and dendritic orientation.

## *Hox5* genes shape the pattern of phrenic motor neuron firing

Our in vivo plethysmography data (*Figure 1*) provided an overview of how the entire respiratory system, including sensory feedback and possible compensatory changes due to hypoxia and hypercapnia, responds to *Hox5* gene deletions. To determine whether the loss of *Hox5*-dependent transcriptional programs specifically affects the activity of phrenic MNs in response to circuitry intrinsic to the brainstem and spinal cord, we performed suction electrode phrenic nerve recordings from isolated brainstem-spinal cord preparations that exhibit fictive breathing (*Figure 6a*) (*Cregg et al., 2017*). We examined whether loss of *Hox5* genes results in changes in phrenic MN activity at P4, a timepoint at which respiratory bursts are especially rhythmic and robust. Despite the changes in PMC clustering and dendritic topography, no significant changes in phrenic nerve burst frequency, rhythmicity, or duration were observed in $Hoxa5^{MN\Delta}$; $c5^{het}$ mice (*Figure 6b–c*), indicating that the brainstem circuits that drive inspiratory bursts, located within the pre-Bötzinger complex, are intact and able to transmit excitatory drive to phrenic MNs via the rVRG.

Our results indicate that parameters that reflect inspiratory/expiratory balance on a long time scale, such as frequency and burst duration, are largely Hox5-independent. However, respiratory efficiency also relies on MN activity on a shorter time scale (i.e. 10–100 ms), and the precise temporal pattern of phrenic MN firing during inspiratory bursts is necessary for forceful diaphragm contractions (*van Lunteren and Sankey, 2000*). We found that MN bursts from control (*Hoxa5 flox/flox; Hoxc5+/-*) mice exhibited a highly reproducible firing pattern, with inspiratory bursts comprised of periods of activity interspersed with periods with no unit activity (*Figure 6d*). These silent periods progressively lengthen throughout the burst. In addition, periods of activity are dominated by large amplitude rhythmic oscillations that occur at approximately 30 Hz (*Figure 6d*), generated by the summations of multiple phrenic MN units firing in the same temporal pattern. Interestingly, these 30 Hz oscillations are matched to the fusion frequency of the diaphragm muscle, which is the frequency of firing at which the diaphragm is tonically and maximally contracted (*Martin-Caraballo et al., 2000*). Thus, patterned firing of phrenic MNs close to the fusion frequency promotes highly efficient diaphragm contraction.

Remarkably, this pattern in inspiratory bursts is abolished in $Hoxa5^{MN\Delta}$; $c5^{het}$ mice (*Figure 6d*). Inspiratory bursts exhibit near continuous firing after loss of *Hox5* genes, and thus the silent periods within the burst are largely lost (*Figure 6e*). In addition, the large amplitude rhythmic 30 Hz activity was eliminated, suggesting that the firing of phrenic MNs in $Hoxa5^{MN\Delta}$; $c5^{het}$ mice is no longer constrained to occur at specific times but is instead distributed throughout the burst, thus reducing the compound action potentials seen in control mice. Power spectrum analysis to resolve the recording into its component frequencies indicated a decrease in 30 Hz activity with a concomitant broad spectrum increase in higher frequencies (*Figure 6f–g*). Importantly, firing of phrenic MNs in an unpatterned manner that does not correlate with the diaphragm fusion frequency provides no additional contractile benefit, and in fact may increase the risk of phrenic MN adaptation, diaphragm muscle fatigue, and respiratory failure (*Martin-Caraballo et al., 2000*). We confirmed these changes were not due to a reduction in phrenic MN number, as blocking apoptosis by introducing the *Bax* deletion into $Hoxa5^{MN\Delta}$; $c5^{het}$ mice did not rescue the phrenic MN firing pattern (*Figure 6—figure supplement 1a–d*). In addition, bursts from $Hoxa5^{MN\Delta}$; $c5^{het}$ mice at the time of birth (P0) display a similar change in motor output (*Figure 6—figure supplement 2a–d*), suggesting that *Hox5* genes function during embryonic development to shape phrenic MN activity at birth. Collectively, these data show that Hox5-dependent transcriptional programs are required for shaping the pattern of phrenic MN output and confining firing to frequency oscillations that promote efficient diaphragm contraction while preventing muscle failure.

## Hox5-dependent PMC inhibition patterns phrenic MN activity

Phrenic MN firing pattern and synchronicity are thought to be generated in part by inhibitory synaptic transmission that modulates firing in response to excitatory drive (*Bou-Flores and Berger, 2001; Marchenko and Rogers, 2009*). We therefore sought to determine whether a loss of inhibitory synaptic transmission underlies the changes in firing present in $Hoxa5^{MN\Delta}$; $c5^{het}$ mice. We performed

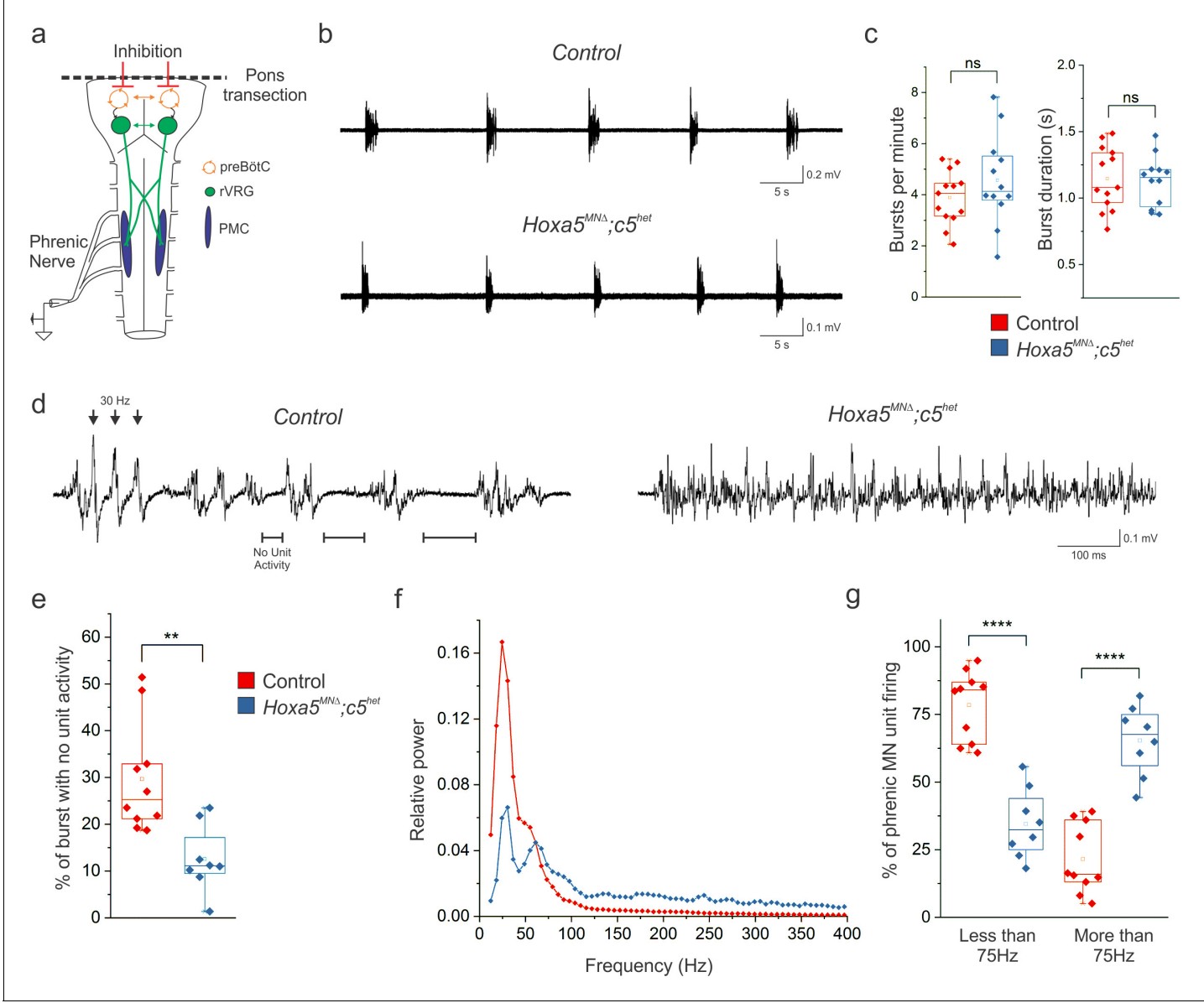

**Figure 6.** *Hox5* genes shape the pattern of phrenic motor neuron activity. (**a**) Schematic of brainstem-spinal cord preparation, which displays fictive inspiration after removal of the pons. Suction electrode recordings were made from the phrenic nerve in the thoracic cavity at P4. (**b–c**) Respiratory burst frequency and burst duration are unaffected in *Hoxa5^MNΔ^; c5^het^* mice. While there is an increase in the variability of burst frequency in *Hoxa5^MNΔ^; c5^het^* mice, it is not statistically significant (n = 13 control, 12 *Hoxa5^MNΔ^; c5^het^*). (**d**) Phrenic firing pattern within inspiratory bursts is dramatically altered in *Hoxa5^MNΔ^; c5^het^* mice. In control mice, bursts are composed of periods of activity interspersed with silent periods with no unit activity (see horizontal lines). Populations of phrenic MNs fire in a pattern with a periodicity of about 30 Hz, where simultaneous firing of units generates large amplitude synchronous compound action potentials (see arrowheads). Phrenic MN firing lacks silent periods and is no longer constrained to certain frequencies in *Hoxa5^MNΔ^; c5^het^* mice. (**e**) The percentage of time during inspiratory bursts with no unit activity is decreased in *Hoxa5^MNΔ^; c5^het^* mice. (**f**) Power spectra showing that control mice have a prominent peak at 30 Hz with little activity above 75 Hz. *Hoxa5^MNΔ^; c5^het^* mice have a decreased 30 Hz peak and broadly increased activity above 75 Hz. For relative power definition, see Materials and methods. (**g**) Percentage of PMC unit firing above and below 75 Hz in control and *Hoxa5^MNΔ^; c5^het^* mice (n = 10 control, 8 *Hoxa5^MNΔ^; c5^het^* for e-g). See also *Figure 6—figure supplements 1* and *2*.
The online version of this article includes the following figure supplement(s) for figure 6:

**Figure supplement 1.** *Hox5* genes shape PMC activity independently of MN survival.
**Figure supplement 2.** *Hox5* genes establish the pattern of phrenic MN firing prior to birth.

unilateral local microinjections of the GABA$_A$ receptor antagonist picrotoxin and the glycine receptor antagonist strychnine into the ventral spinal cord at C3-C6, the location of the PMC. Injection of picrotoxin and strychnine in control mice resulted in a firing pattern indistinguishable from that seen in *Hoxa5$^{MN\Delta}$; c5$^{het}$* mice (**Figure 7a**). Disinhibited control phrenic MNs fired throughout the burst with reduced periods of no activity (**Figure 7b**). Power spectra analysis revealed a decrease in 30 Hz activity with a concomitant broad increase in high frequency activity (**Figure 7c–d**). Picrotoxin and strychnine injections into *Hoxa5$^{MN\Delta}$; c5$^{het}$* mice had little effect on phrenic MN firing (**Figure 7a**).

The ability to convert the firing pattern of control mice into one similar to *Hoxa5$^{MN\Delta}$; c5$^{het}$* mice by disinhibition locally on the PMC, and the fact that disinhibition had little effect on *Hoxa5$^{MN\Delta}$; c5$^{het}$* phrenic MN firing, implies that *Hoxa5$^{MN\Delta}$; c5$^{het}$* mice have lost the inhibitory synaptic transmission which is important for generating this pattern. We explored whether we could detect any anatomical alterations in inhibitory synaptic inputs by performing synaptic puncta quantitation. We counted perisomatic inhibitory synapses, as defined by apposition of the presynaptic marker GAD67 and the postsynaptic marker gephyrin, on phrenic MNs in control and *Hoxa5$^{MN\Delta}$; c5$^{het}$* mice (**Figure 7e–f**). *Hoxa5$^{MN\Delta}$; c5$^{het}$* mice exhibited a 20% reduction in inhibitory synapse number as compared to control (**Figure 7g**). Our results show that functional phrenic MN output is altered in the absence of *Hox5* genes, likely due to loss of a subset of inhibitory inputs that act to pattern motor output. Together, our data support a model where Hox5-dependent transcriptional programs shape the pattern of respiratory output by establishing inhibitory inputs onto phrenic MNs.

## Discussion

Phrenic MNs are the final output of complex respiratory circuits and provide motor drive to the diaphragm, the major inspiratory muscle in mammals. Despite their critical function, the molecular mechanisms that control their selective targeting by premotor populations and shape their output are largely unknown. In this study, we find that Hox5 transcription factors control phrenic MN topography by establishing a PMC-specific cadherin code. MN-specific *Hox5* deletion impacts the formation of inhibitory inputs onto PMC neurons, alters phrenic MN output, and leads to respiratory dysfunction throughout life. We discuss these findings in the context of respiratory circuit assembly and breathing behaviors.

### *Hox5* genes control PMC clustering and topography through a phrenic-specific cell adhesion code

The establishment of clustered neuronal nuclei is a conserved and prominent organizational feature of the CNS and is thought to be critical for neural circuit assembly (*Jessell et al., 2011*). Phrenic MN clustering serves an additional function. The generation of fetal breathing movements is required for lung and diaphragm maturation; however, descending inspiratory drive is relatively weak during embryonic development (*Greer, 2012*). Electrical coupling between tightly clustered phrenic MNs facilitates the recruitment of multiple motor units to compensate for weak inputs and generate adequate synchronous motor drive to the diaphragm (*Greer and Funk, 2005*). While electrical coupling is not observed in mature phrenic MNs, when maximum motor unit recruitment becomes inefficient and is no longer desirable, it is especially beneficial during the development of the respiratory system (*Greer et al., 1999*). Therefore, phrenic MN clustering likely serves a dual function during development, enabling both electrical coupling and premotor targeting. Our results indicate that Hox5 transcription factors regulate a molecular program that defines PMC position and clustering.

Our RNA-seq analysis revealed a number of cell adhesion molecules that are downregulated in the absence of *Hox5* genes. CAM expression also appears to distinguish phrenic MNs from other MN populations, alluding to MN subtype-specific CAM functions (*Machado et al., 2014*). Comprehensive analysis of cadherin expression identified a combinatorial PMC cadherin code (cdh2, 6, 9, 10, 11 and 22). Since our strategy of inactivating *β/γ-catenin* in MNs eliminated all cadherin signaling, we cannot definitively establish whether the full complement of PMC-specific cadherins is necessary for efficient phrenic MN clustering. Expression of single or a small subset of cadherins may be insufficient to endow PMC neurons with a unique identity for self-recognition, and multiple cadherin expression might be necessary for their segregation from limb-innervating and axial MNs found at the same levels. PMC cadherins belong to all three specificity groups identified recently on the basis of their heterophilic interactions, further restricting the likelihood that PMC neurons will interact with

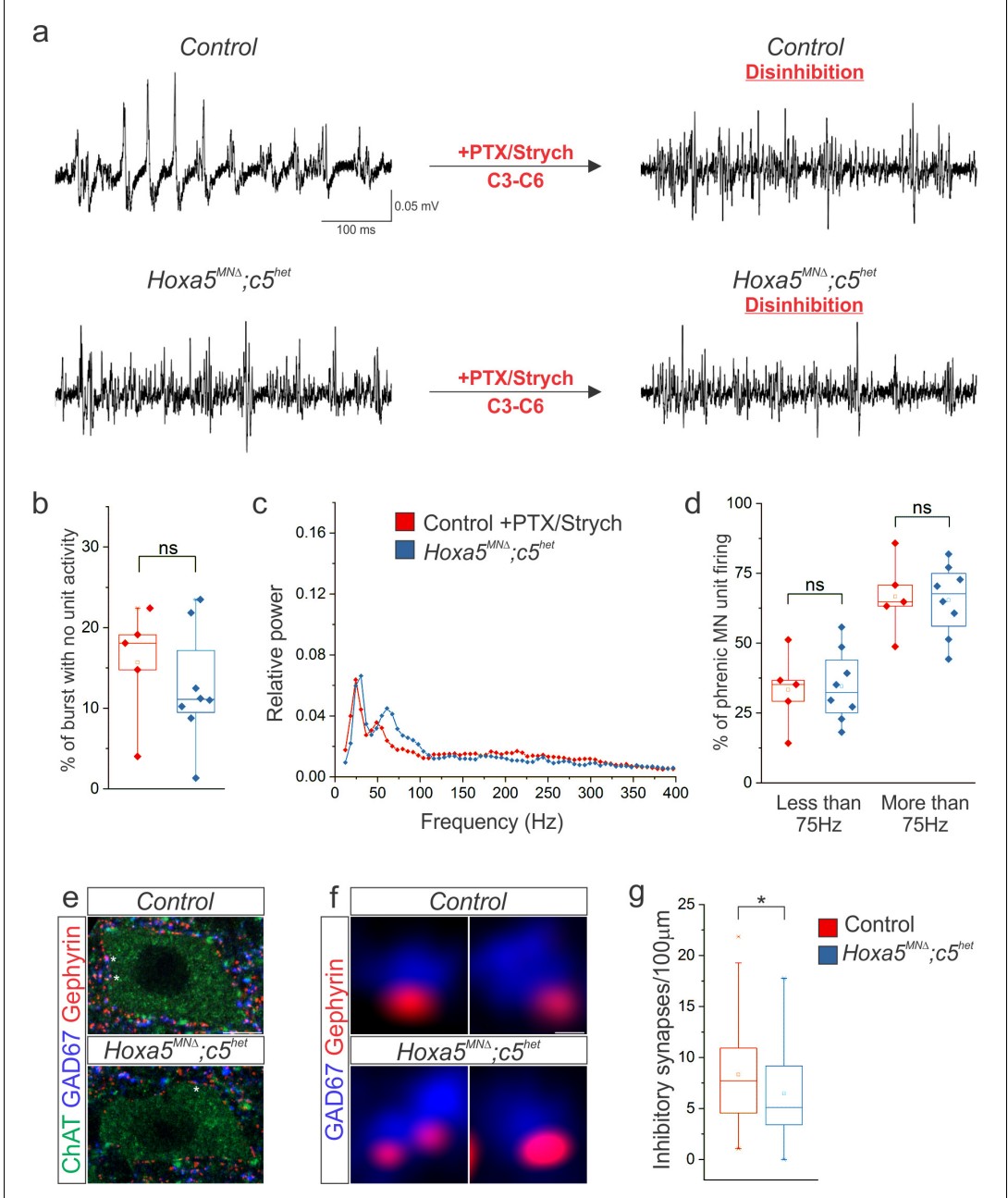

**Figure 7.** Hox5-dependent PMC inhibition patterns PMC activity. (**a**) Disinhibition via local microinjection of picrotoxin (GABA_A receptor antagonist) and strychnine (glycine receptor antagonist) into the ventral spinal cord at C3-C6, the location of the PMC, converts the highly patterned firing of phrenic MNs in control mice to the more continuous firing observed in *Hoxa5^{MNΔ}; c5^{het}* mice. (**b**) The percentage of time during inspiratory bursts with no unit activity is equivalent in both control +PTX/Strych and *Hoxa5^{MNΔ}; c5^{het}* mice, and is considerably reduced from control mice (*Figure 6*). (**c–d**) Power spectra and the percentage of phrenic MN unit firing above and below 75 Hz are equivalent in control +PTX/Strych and *Hoxa5^{MNΔ}; c5^{het}* mice. For relative power definition, see Materials and methods (n = 5 control +PTX/Strych, 8 *Hoxa5^{MNΔ}; c5^{het}* for b-d). (**e**) The number of perisomatic inhibitory synapses on phrenic MNs, identified by the apposition of GAD67 and gephyrin, is reduced in P10 *Hoxa5^{MNΔ}; c5^{het}* mice. A single phrenic MN cell body is shown and representative inhibitory synapses are labelled with stars. Choline Acetyltransferase (ChAT) staining was used to identify the cell body membrane. (**f**) Examples of inhibitory perisomatic synapses showing apposition of GAD67 and gephyrin. (**g**) Quantification of the reduction in perisomatic inhibitory synapses in *Hoxa5^{MNΔ}; c5^{het}* mice (n = 86 phrenic MNs from 2 control mice, 78 phrenic MNs from 2 *Hoxa5^{MNΔ}; c5^{het}* mice). Scale bar = 10 μm in e, 0.25 μm in f. See also *Figure 7—figure supplement 1*.

The online version of this article includes the following figure supplement(s) for figure 7:

**Figure supplement 1.** *Hox5* genes control PMC topography and connectivity.

other MN populations through heterophilic interactions between cadherins of the same subgroup (*Brasch et al., 2018*). While the function of the type I cadherin, N-cadherin is required for clustering of all MN populations, additional type II cadherin divergent expression among MN populations may be required for specific MN subtype clustering (*Dewitz et al., 2019*).

We also demonstrate that *Hox5* genes are required for the correct orientation of PMC dendrites. Phrenic MN dendrites form tightly fasciculated bundles that adopt a distinct ventromedial and dorsolateral orientation during development (*Allan and Greer, 1997*). By late embryonic stages (e18.5), phrenic dendrites are restricted to the ipsilateral side and rarely cross the midline. Loss of *Hox5* genes leads to a loss of stereotypical dendritic organization and an increased crossing to the contralateral site. Loss of cadherin function leads to similar dendritic reorganization and loss of dorsolateral dendrites, arguing that cadherins are key effectors of Hox5 action. However, cadherin loss does not increase the number of dendrites that cross the midline, pointing to additional mechanisms acting downstream of Hox5 proteins. Notably, while cadherin signaling is critical for dendritic growth and orientation, $\beta/\gamma$-catenin inactivation does not impair phrenic MN axon growth or diaphragm innervation, suggesting that *Hox5* genes control two independent molecular programs that regulate axonal and dendritic growth respectively, to coordinate the wiring specificity of phrenic MNs. The requirement for cadherins in establishing both cell body and dendritic topography suggests a prominent cadherin role in PMC presynaptic connectivity. Interestingly, a subpopulation of neurons in the pre-Bötzinger complex also expresses cdh9, which could indicate a broad role for cadherins is establishing synaptic connectivity throughout respiratory circuits (*Yackle et al., 2017*).

## MN identity and the logic of selective presynaptic connectivity

How are phrenic MNs specifically targeted by respiratory premotor populations while eschewing inputs from other descending neurons, locomotor-related interneurons, and sensory afferents directed to other nearby MN populations? MN identity, established by early transcriptional programs, is emerging as a critical determinant of MN connectivity (*Dasen, 2017*). Despite eradicating phrenic MN identity through *Hox5* deletion, which led to loss of PMC topography and downregulation of PMC-specific CAMs, descending excitatory inputs to PMC neurons appear to be largely unperturbed, as we still observe regular phrenic MN bursting in isolated brainstem-spinal preparations. The persistence of excitatory inputs could indicate that Hox5-dependent precise PMC topography is not necessary for these populations to synapse on phrenic MNs or that multiple redundant mechanisms have evolved to maintain this connection that is extremely critical for life. Loss of *Hox5* genes appears to selectively impact the establishment of inhibitory inputs onto phrenic MNs, suggesting distinct requirements for PMC targeting by individual premotor populations.

How do *Hox5* genes influence PMC connectivity? In sensory-motor circuits, the correct positioning of MNs appears to be critical for their targeting by sensory axons (*Sürmeli et al., 2011*). In addition, it has recently been reported that spatial features of the MN dendritic tree, such as the angle of interaction with approaching axons, can also act as a determinant of their connectivity with sensory neurons (*Balaskas et al., 2019*). Here, we demonstrate that *Hox5* genes determine PMC cell body and dendritic topography through the induction of cadherin expression. Do cadherins solely function to position phrenic MNs and dendrites at the right place during development or do they have additional roles as molecular recognition cues in presynaptic targeting? In the retina, cadherins control the topography of axonal and dendritic arbors of synaptic partners to place them in close proximity and enable synaptogenesis (*Duan et al., 2014*; *Duan et al., 2018*). In the hippocampus however, cadherins influence synaptic fidelity and potentiation without overtly affecting cell morphology, pointing to position-independent roles in synaptic connectivity (*Basu et al., 2017*). Similarly, mutations in transcription factors that alter molecular identity but not cell body position dramatically reconfigure MN inputs, indicating that MN position is unlikely to be the only critical parameter for MN connectivity (*Hinckley et al., 2015*; *Machado et al., 2015*). Loss of inhibitory inputs onto PMC neurons likely results from both positional and molecular changes resulting from MN-specific *Hox5* deletion. Defining the explicit contribution of topography to phrenic MN connectivity will require decoupling PMC position from Hox5-mediated transcriptional programs that also control the molecular determinants of phrenic identity.

## Phrenic MN inhibition and respiratory output

The major function of phrenic MNs is to efficiently contract the diaphragm muscle, and as such, MNs could potentially function to merely execute complex computations occurring in upstream brainstem respiratory circuits. Consistent with this idea, recent monosynaptic viral-based retrograde tracing of phrenic MN inputs revealed that the major PMC projection arises from excitatory rVRG neurons that propagate the respiratory rhythm generated by the pre-Bötzinger complex (*Wu et al., 2017*). However, in addition to this excitation, phrenic MNs receive multiple modulatory inputs, including serotonergic and cholinergic inputs, indicating at least some degree of computation transforms rhythmic signals into appropriate motor patterns at the MN level. Phrenic MNs also receive substantial descending inhibitory inputs and we observed an abundance of inhibitory synapses on phrenic MN cell bodies. While we did not observe a complete loss of inhibitory synapses in $Hoxa5^{MN\Delta}$; $c5^{het}$ mice, it is likely that only a subset of phrenic MN inhibition is dedicated to patterning motor neuron activity within inspiratory bursts. Alternatively, while we still observe synaptic puncta on phrenic cell bodies, a number of these synapses may be non-functional, as cadherins are also required for synaptic organization (*Yamagata et al., 2018*).

What is the source of this inhibition and how does it contribute to shaping phrenic MN output? Rabies-virus mediated retrograde tracing revealed a population of PMC-projecting inhibitory neurons within the rVRG (*Wu et al., 2017*). Excitatory and inhibitory rVRG neurons are activated concurrently, such that inhibition is in phase with excitatory inputs generating inspiration (*Parkis et al., 1999*). This inspiratory-phase inhibition synchronizes MN output on a short time scale, and this oscillation frequency is thought to match the frequency that produces maximal muscle force to generate robust diaphragm contractions (*Bou-Flores and Berger, 2001*; *Parkis et al., 2003*; *Sebe et al., 2006*). Our results demonstrate that *Hox5* genes are required for establishing these inhibitory inputs onto phrenic MNs, revealing how early transcriptional programs contribute to phrenic MN patterned output.

## *Hox5* genes and respiratory dysfunction

Loss of *Hox5* genes results in pronounced defects in breathing behaviors, including reductions in tidal volume and inability to respond to respiratory challenges such as hypercapnia. There are likely multiple contributing factors to this respiratory dysfunction, including the loss of diaphragm innervation, the reduction in phrenic MN numbers and the inefficient activation of the remaining phrenic MNs due to the loss of electrical coupling and inhibitory inputs (*Figure 7—figure supplement 1*). Differences in tidal volume are somewhat mitigated with age, reflecting the decrease in motor unit recruitment during quiet breathing with the maturation of the respiratory system (*Greer et al., 1999*). At birth, the vast majority of phrenic MNs are recruited at rest, making the impact of phrenic MN loss on tidal volume more pronounced. As motor units mature, recruitment of a small subset of phrenic MNs is sufficient to generate efficient diaphragm contractions, partly compensating for phrenic MN loss. However, upon a respiratory challenge such as hypercapnia, no additional motor units are available to be recruited in $Hoxa5^{MN\Delta}$; $c5^{het}$ mice, leading to severe decreases in minute ventilation and compromising the hypercapnia response.

In $Hoxa5^{MN\Delta}$; $c5^{het}$ mice there is a gradual compensation for tidal volume reduction by increasing breathing frequency at rest. However, this compensation does not occur until 2 weeks of age, likely reflecting the maturation of central chemosensory regions that provide respiratory feedback (*Putnam et al., 2005*). This indicates that *Hox5* mutations render mice particularly vulnerable to respiratory failure during the first 2 weeks of life, and consistent with this we observe increased perinatal lethality in $Hoxa5^{MN\Delta}$; $c5^{het}$ mice, reminiscent of SIDS. While much attention has been focused on identifying deleterious genetic variants that impair $CO_2$-sensing populations, such as serotonergic neurons, as causal to SIDS, another possibility is that gene variants causing defects in phrenic MN connectivity and function may be an alternative risk factor for neonatal lethality (*Kinney et al., 2009*; *Rand et al., 2013*; *Van Norstrand and Ackerman, 2010*). As GWAS studies are becoming increasingly common, they may in the future reveal that mutations in *Hox5* transcription factors, or downstream cell adhesion molecules, also lead to respiratory dysfunction in humans and are a genetic risk factor for SIDS.

# Materials and methods

## Key resources table

| Reagent type (species) or resource | Designation | Source or reference | Identifiers | Additional information |
|---|---|---|---|---|
| Gene (*M. musculus*) | *Hoxa5* | | MGI:96177 | |
| Gene (*M. musculus*) | *Hoxc5* | | MGI:96196 | |
| Gene (*M. musculus*) | *Alcam* | | MGI:1313266 | |
| Gene (*M. musculus*) | *Edil3* | | MGI:1329025 | |
| Gene (*M. musculus*) | *Ptprt* | | MGI:1321152 | |
| Gene (*M. musculus*) | *Lsamp* | | MGI:1261760 | |
| Gene (*M. musculus*) | *Negr1* | | MGI:2444846 | |
| Gene (*M. musculus*) | *Cdh2* | | MGI:88355 | |
| Gene (*M. musculus*) | *Cdh6* | | MGI:107435 | |
| Gene (*M. musculus*) | *Cdh9* | | MGI:107433 | |
| Gene (*M. musculus*) | *Cdh10* | | MGI:107436 | |
| Gene (*M. musculus*) | *Cdh11* | | MGI:99217 | |
| Gene (*M. musculus*) | *Cdh22* | | MGI:1341843 | |
| Genetic reagent (*M. musculus*) | *Hoxa5 flox* | PMID: 17417799 | MGI:3723622 | |
| Genetic reagent (*M. musculus*) | *Hoxc5-/-* | PMID: 17626057 | MGI:3526151 | |
| Genetic reagent (*M. musculus*) | *Olig2::Cre* | PMID: 18046410 | MGI:3774124 | |
| Genetic reagent (*M. musculus*) | *Hb9::GFP* | PMID: 12176325 | MGI:3056906; RRID:IMSR_JAX:005029 | |
| Genetic reagent (*M. musculus*) | *Bax-/-* | PMID: 7569956 | MGI:1857429; RRID:IMSR_JAX:002994 | |
| Genetic reagent (*M. musculus*) | *β-catenin flox* | PMID: 11262227 | MGI:2148567; RRID:IMSR_JAX:004152 | |
| Genetic reagent (*M. musculus*) | *γ-catenin flox* | PMID: 22036570 | MGI:5305426 | |
| Genetic reagent (*M. musculus*) | *γ-catenin-/-* | PMID: 8858175 | MGI:1861958; RRID:IMSR_JAX:003334 | |
| Antibody | anti-scip (goat polyclonal) | Santa Cruz Biotechnology | RRID:AB_2268536 | (1:5000) |
| Antibody | anti-islet1/2 (mouse monoclonal) | DSHB; PMID: 7528105 | RRID:AB_2314683 | (1:1000) |
| Antibody | anti-neurofilament (rabbit polyclonal) | Synaptic Systems | RRID:AB_887743 | (1:1000) |
| Antibody | anti-synaptophysin (rabbit monoclonal) | Thermo Fisher | RRID:AB_10983675 | (1:250) |
| Antibody | anti-GFP (rabbit polyclonal) | Invitrogen | RRID:AB_221570 | (1:1000) |
| Antibody | anti-ChAT (goat polyclonal) | Millipore | RRID:AB_2079751 | (1:200) |
| Antibody | anti-GAD67 (mouse monoclonal) | Millipore | RRID:AB_2278725 | (1:500) |
| Antibody | anti-gephyrin (mouse monoclonal) | Synaptic Systems | RRID:AB_2232546 | (1:3000) |
| Commercial assay or kit | TSA Amplification Kit | Perkin Elmer | Cat. No. NEL753001KT | |

*Continued on next page*

*Continued*

| Reagent type (species) or resource | Designation | Source or reference | Identifiers | Additional information |
|---|---|---|---|---|
| Commercial assay or kit | Arcturus Picopure RNA Isolation Kit | Applied Biosystems | Cat. No. KIT0204 | |
| Commercial assay or kit | Stranded RNA-seq Kit with Riboerase | Kapa Biosystems | Cat. No. KK8483 | |
| Chemical compound, drug | Picrotoxin | Tocris Bioscience | Cat. No. 1128 | 1.25 mM |
| Chemical compound, drug | Strychnine hydrochloride | Sigma | Cat. No. S8753 | 1.25 mM |
| Chemical compound, drug | α-bungarotoxin, Alexa Fluor 555 conjugate | Invitrogen | RRID:AB_2617152 | (1:1000) |
| Chemical compound, drug | diI | Invitrogen | Cat. No. D3911 | |
| Software, algorithm | FASTQC | http://www.bioinformatics.babraham.ac.uk/projects/fastqc/ | RRID:SCR_014583 | |
| Software, algorithm | Trim Galore | http://www.bioinformatics.babraham.ac.uk/projects/trim_galore/ | RRID:SCR_011847 | |
| Software, algorithm | Tophat v2.1.0 | PMID: 23618408 | RRID:SCR_013035 | |
| Software, algorithm | htseq-count | PMID: 25260700 | RRID:SCR_011867 | |
| Software, algorithm | R v3.6.1 | http://www.r-project.org/ | RRID:SCR_001905 | |
| Software, algorithm | pClamp 10 | Molecular Devices | RRID:SCR_011323 | |
| Software, algorithm | Spike2 | Cambridge Electronic Design | RRID:SCR_000903 | |

## Mouse genetics

The *loxP*-flanked *Hoxa5* (*Tabariès et al., 2007*), *β-catenin* (*Brault et al., 2001*), and *γ-catenin* (*Demireva et al., 2011*) alleles, *Hoxc5* mutant strains (*McIntyre et al., 2007*), *Olig2::cre* (*Dessaud et al., 2007*), *Hb9::GFP* (*Wichterle et al., 2002*), *γ-catenin-/-* (*Ruiz et al., 1996*), and *Bax-/-* (*Knudson et al., 1995*) lines were generated as previously described and maintained on a mixed background. Mouse colony maintenance and handling was performed in compliance with protocols approved by the Institutional Animal Care Use Committee of Case Western Reserve University. Mice were housed in a 12 hr light/dark cycle in cages containing no more than five animals at a time.

## Immunohistochemistry and in situ hybridization

In situ hybridization and immunohistochemistry were performed as described (*Philippidou et al., 2012*) on tissue fixed for 2 hr in 4% PFA and cryosectioned at 16 μm (12 μm for synaptic puncta quantitation). In situ probes were generated from e12.5 cervical spinal cord cDNA libraries using PCR primers with a T7 RNA polymerase promoter sequence at the 5' end of the reverse primer. All probes generated were 600–1000 bp in length. The sequences used for the PCR primers, probe length, and additional BLAST results verifying specificity of the cadherin probes is located in the attached *Supplementary file 1*. Wholemounts of diaphragm muscles were stained as described (*Philippidou et al., 2012*). Diaphragm staining was performed using either neurofilament/synaptophysin primary antibodies (for mice without *Hb9::GFP*) or with GFP primary antibodies (for mice with *Hb9::GFP*). The following antibodies were used: goat anti-Scip (1:5000; Santa Cruz Biotechnology, RRID:AB_2268536), mouse anti-islet1/2 (1:1000, DSHB, RRID:AB_2314683) (*Tsuchida et al., 1994*), rabbit anti-neurofilament (1:1000; Synaptic Systems, RRID:AB_887743), rabbit anti-synaptophysin (1:250, Thermo Fisher, RRID:AB_10983675), rabbit anti-GFP (1:1000, Invitrogen, RRID:AB_221570), α-bungarotoxin, Alexa Fluor 555 conjugate (1:1000; Invitrogen, RRID:AB_2617152), goat anti-ChAT (1:200, Millipore, RRID:AB_2079751), mouse anti-GAD67 (1:500, Millipore, RRID:AB_2278725), and mouse anti-gephyrin (1:3000, Synaptic Systems, RRID:AB_2232546). Images were obtained with a Zeiss LSM 800 confocal microscope and analyzed with Zen Blue and ImageJ (Fiji). Diaphragm

innervation was quantified using the simple neurite tracer plugin in ImageJ. For synaptic puncta quantitation performed at P10, phrenic motor neurons were identified by determining the proper rostrocaudal level using surrounding Hoxa5 and Hoxc8 expression. High resolution individual synaptic puncta were imaged using Zeiss Airyscan.

## DiI tracing

For labeling of phrenic motor neurons, crystals of carbocyanine dye, DiI (Invitrogen, #D3911) were pressed onto the phrenic nerves of eviscerated embryos, and the embryos were incubated in 4% paraformaldehyde at 37°C in the dark for 2 weeks for e14.5 embryos and 4–5 weeks for e18.5 embryos. Subsequently, spinal cords were dissected, embedded in 4% low melting point agarose (Invitrogen) and sectioned using a Leica VT1000S vibratome at 100 to 150 μm.

## Cell body clustering and dendritic orientation analysis

To calculate the clustering index for PMC neurons, we utilized two complementary approaches. For experiments with membrane staining (diI tracing), the number of retrogradely-traced neurons that were in contact with at least one other labelled neuron was counted and divided by the total number of traced neurons to calculate a clustering index. A clustering index of 1 indicates that all retrogradely-traced MNs were clustered. For experiments with nuclear protein staining, we connected all PMC nuclei (Scip+Isl1/2+) to their nearest neighbor to form a perimeter of the PMC. We then counted the number of phrenic MNs and the area occupied by the PMC. Clustering index was defined as the number of PMC neurons/1000 μm$^2$.

For the analysis of dendritic orientation in $Hoxa5^{MN\Delta}$; $c5^{het}$ mice, we superimposed a grid over phrenic MN cell bodies spanning 200 μm in each direction. We then used Fiji (ImageJ) to calculate the fluorescent intensity in each square and divided this by the sum of the total fluorescent intensity to calculate the percentage of dendrites in each area. For $\beta\gamma$-$cat^{MN\Delta}$ mice, since MN cell bodies were more dispersed, we only analyzed dorsal dendrites by calculating the fluorescent intensity of dendrites projecting dorsal to cell bodies divided by the total intensity.

## RNA-sequencing and analysis

C3-C6 cervical spinal cords were dissected from 3 control (Bax-/-) and 3 $Hox5^{MN\Delta}$; Bax-/- mutant embryos in a Hb9::GFP background at e12.5 and motor neurons were sorted by fluorescence–activated cell sorting (FACS) on a Sony iCyt cell sorter. RNA was extracted using the PicoPure RNA isolation system (Arcturus, #KIT0204) with RIN >8 via Tapestation analysis (Agilent). rRNA depleted libraries were prepared from 10 to 20 ng of total RNA using the KAPA stranded RNA-seq kit with Riboerase (KAPA, #KK8483) and amplified by 15 cycles of PCR. Paired-end, 150 bp sequencing was performed on the Illumina HiSeq 2500 and generated a total of 58–94 million reads in each direction per sample after filtering. Read quality was assessed using FASTQC (https://www.bioinformatics. babraham.ac.uk/projects/fastqc/) and adapters were trimmed using Trim Galore (https://www.bioin-formatics.babraham.ac.uk/projects/trim_galore/). Filtered and trimmed reads were aligned to the mouse genome (GRCm38.p5) using TopHat v2.1.0 (*Kim et al., 2013*). Gene counts were obtained using htseq-count (*Anders et al., 2015*). Differentially expressed genes were considered significant if p<0.05 due to the dilution factor from the lack of genetic tools available to specifically sort PMC neurons. Plots were created in R (v 3.5). The R package pheatmap was used to generate the hierarchical clustered heatmap using row-scaled values of differentially expressed genes with p<0.05. Function enrichment was performed using the R package gProfileR with FDR < 0.05. The resulting GO terms were simplified based on similarity using REVIGO (*Supek et al., 2011*).

## Electrophysiology

Electrophysiology was performed as previously described (*Cregg et al., 2017*). Mice were cryoanesthetized and rapid dissection was carried out in 22–25°C oxygenated Ringer's solution. The solution was composed of 128 mM NaCl, 4 mM KCl, 21 mM NaHCO$_3$, 0.5 mM NaH$_2$PO$_4$, 2 mM CaCl$_2$, 1 mM MgCl$_2$, and 30 mM D-glucose and was equilibrated by bubbling in 95% O$_2$/5% CO$_2$. The hindbrain and spinal cord were exposed by ventral laminectomy, and phrenic nerves exposed and dissected free of connective tissue. A transection at the pontomedullary boundary rostral to the anterior inferior cerebellar artery was used to initiate fictive inspiration. Electrophysiology was performed under

continuous perfusion of oxygenated Ringer's solution in a rostral to caudal direction to prevent drug diffusion to the brainstem during local injection. Suction electrodes were attached to phrenic nerves just proximal to their arrival at the diaphragm. For local injections, we used the following drugs: picrotoxin (PTX) ($GABA_A$ receptor antagonist, 1.25 mM, Tocris Bioscience, #1128) and strychnine hydrochloride (Strych) (glycine receptor antagonist, 1.25 mM, Sigma, #S8753) dissolved in Ringer's solution with trypan blue for visualization. The signal was band-pass filtered from 10 Hz to 3 kHz using Grass amplifiers, amplified 5,000-fold, and sampled at a rate of 50 kHz with a Digidata 1440A (Molecular Devices). Data were recorded using AxoScope software (Molecular Devices) and analyzed in Spike2 (Cambridge Electronic Design). Burst duration, percent of burst time with no activity, and power spectra were computed from five bursts per mouse, while burst frequency was determined from 5 min of recording time per mouse. Traces analyzed for percent of burst time with no activity and power spectral analysis were of similar amplitude. In injection experiments, bursts were analyzed 8–10 min after injection. For power spectra, relative power is defined as the absolute power for that frequency bin divided by the sum of the absolute power over all frequency bins from 10 Hz – 400 Hz. Control mice for electrophysiology experiments were all *Hoxa5 flox/flox; Hoxc5+/-.*

## Plethysmography

Conscious, unrestrained adult (P80) mice were placed in a whole body, flow through plethysmograph chamber (emka) attached to a differential pressure transducer (emka). The chamber was filled with normal air (79% nitrogen, 21% oxygen), and flow was maintained at 0.75 L/min per chamber for all gas mixtures. Mice were acclimated for 1 hr in normal air, and then a 5% $CO_2$ mixture (74% nitrogen, 21% oxygen, 5% carbon dioxide) was introduced to the chamber for 15 min, after which the mice were removed. Experiments were performed in pairs, with each pair consisting of one littermate control and one experimental mouse of the same sex and approximate weight. Thirty seconds of resting breaths were analyzed using iox2 software (emka) near the end of the acclimation period in normal air, and another thirty seconds of resting breaths were analyzed 10 to 15 min after the introduction of 5% $CO_2$. Mice were directly observed to identify resting breaths. Each mouse was recorded on three consecutive days and the values were averaged together to reduce variability. Data are presented as fold control, where the control is the matched littermate in normal air. For neonatal plethysmography, we modified syringes to use as chambers, as smaller chambers increase signal detection. Littermate pups were recorded in normal air every 3–4 days from P3 to P17, and 30–50 breaths were analyzed. Control mice for plethysmography experiments were all *Hoxa5 flox/flox; Hoxc5+/-.*

## Experimental design and statistical analysis

For all experiments a minimum of three embryos per genotype, both male and female, were used for all reported results unless otherwise stated. Genotypes for control mice in Hox5 experiments include cre negative *Hoxa5 flox/flox; Hoxc5+/-* and *Hoxa5 flox/flox; Hoxc5+/+* mice. Genotypes for control mice in catenin experiments include cre negative *β-catenin flox/flox; γ-catenin flox/-* and *β-catenin flox/flox; γ-catenin flox/+* mice. Data are presented as box and whisker plots with each dot representing data from one mouse unless otherwise stated. Small open squares in box and whisker plots represent the mean. P-values were calculated using unpaired, two-tailed Student's *t* test. $p < 0.05$ was considered to be statistically significant, where *$p < 0.05$, **$p < 0.01$, ***$p < 0.001$, and ****$p < 0.0001$.

## Accession numbers

The RNA-seq data reported in this paper is uploaded to the NCBI Gene Expression Omnibus under accession number GSE138085.

## Acknowledgements

We thank Heather Broihier, Evan Deneris, Ashleigh Schaffer, Helen Miranda, Jerry Silver and members of the Philippidou lab for helpful discussions and comments on the manuscript, and James Ferguson for his assistance with RNA-seq analysis. We thank Catarina Catela for advice on fluorescence in situ hybridization. We also thank the Cytometry and Imaging core at Case Comprehensive Cancer

Center (P30CA043703) for assistance with FACS and Admera Health for assistance performing RNA-sequencing.

## Additional information

### Funding

| Funder | Grant reference number | Author |
|---|---|---|
| National Institute of Neurological Disorders and Stroke | R00NS085037 | Polyxeni Philippidou |
| Mt Sinai Health Foundation | | Polyxeni Philippidou |
| Eunice Kennedy Shriver National Institute of Child Health and Human Development | F30HD096788 | Alicia N Vagnozzi |
| National Institute of General Medical Sciences | T32GM007250 | Alicia N Vagnozzi |
| National Institute of Neurological Disorders and Stroke | R01NS114510 | Polyxeni Philippidou |

The funders had no role in study design, data collection and interpretation, or the decision to submit the work for publication.

### Author contributions

Alicia N Vagnozzi, Lynn T Landmesser, Conceptualization, Investigation, Methodology; Kiran Garg, Matthew T Moore, Investigation; Carola Dewitz, Investigation, Methodology; Jared M Cregg, Methodology; Lucie Jeannotte, Resources; Niccolò Zampieri, Resources, Methodology; Polyxeni Philippidou, Conceptualization, Supervision, Funding acquisition, Investigation, Methodology

### Author ORCIDs

Alicia N Vagnozzi (ID) https://orcid.org/0000-0002-6152-8728
Jared M Cregg (ID) http://orcid.org/0000-0002-0027-9748
Niccolò Zampieri (ID) http://orcid.org/0000-0002-2228-9453
Polyxeni Philippidou (ID) https://orcid.org/0000-0002-0733-3591

### Ethics

Animal experimentation: All animal procedures performed in this study were in strict accordance with the Guide for the Care and Use of Laboratory Animals of the National Institutes of Health. The protocol was approved by the Case Western Reserve University School of Medicine Institutional Animal Care and Use Committee (Animal Welfare Assurance Number A3145-01, protocol #: 2015-0180).

### Decision letter and Author response

Decision letter https://doi.org/10.7554/eLife.52859.sa1
Author response https://doi.org/10.7554/eLife.52859.sa2

## Additional files

### Supplementary files

• Supplementary file 1. In situ hybridization probes. Supplementary Table 1. PCR primers for in situ hybridization. Supplementary Table 2. Specificity of in situ probes for PMC cadherins. To ascertain the specificity for the designed in situ probes to its intended target (given the degree of similarity between members of the cadherin family), we ran each full length probe through BLAST (https://blast.ncbi.nlm.nih.gov/) against the refseq_rna database for *M. musculus* and checked for cross-reactivity with other cadherins. BLAST outputs a 'max alignment score' parameter, which awards points for base matches while penalizing base mismatches, gaps, etc. While all on target alignment scores

are high (>1400, dependent on probe length), off target scores are low (<35% of the max alignment score of the intended target).

- Transparent reporting form

### Data availability

Sequencing data have been deposited in GEO under accession code GSE138085.

The following dataset was generated:

| Author(s) | Year | Dataset title | Dataset URL | Database and Identifier |
|---|---|---|---|---|
| Vagnozzi AN, Garg K, Dewitz C, Moore MT, Cregg JM, Jeannotte L, Zampieri N, Landmesser LT, Philippidou P | 2019 | Gene expression changes in cervical motor neuron transcriptomes after loss of Hox5 transcription factors | https://www.ncbi.nlm.nih.gov/geo/query/acc.cgi?&acc=GSE138085 | NCBI Gene Expression Omnibus, GSE138085 |

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
