## [Decision Letter]

**Acceptance summary:**

In this study the authors address the function of the Hox family of transcription factors in assembly of the motor neuron circuits required for breathing. They perform a detailed characterization in mouse of the perturbed connectivity and neuronal firing that arises from partial loss of Hox function, and they identify cell adhesion molecules as targets of Hox regulation that underlie the development of the proper wiring pattern. This study is particularly significant for the fact that it links molecular mechanisms of development with the function of a physiologically important neuronal circuit.

**Decision letter after peer review:**

Thank you for submitting your article "Phrenic-specific transcriptional programs shape respiratory motor output" for consideration by *eLife*. Your article has been reviewed by three peer reviewers, one of whom is a member of our Board of Reviewing Editors, and the evaluation has been overseen by Ronald Calabrese as the Senior Editor. The following individuals involved in review of your submission have agreed to reveal their identity: Richard S Mann (Reviewer #2).

The reviewers have discussed the reviews with one another and the Reviewing Editor has drafted this decision to help you prepare a revised submission.

Summary:

In this manuscript the authors perform a compound motor neuron knockout of the transcription factor Hoxa5 with a Hoxc5 HET to address the function of Hox TFs in assembly of the motor neuron circuits required for breathing. Leaving one copy of Hoxc5 intact in MNs is compatible with survival. This allows them to study the consequences of Hox5 knockout in MNs on the connectivity, gene expression, and function of these neurons. The authors provide data in four main areas. First, they show evidence of breathing abnormalities and susceptibility to perinatal hypoxia. Second, they examine the morphology of the phrenic motor neurons and show that in addition to impaired innervation of the diaphragm, the cell bodies are less clustered and the dendrites are abnormally distributed. Third the authors do RNAseq on purified motor neurons, reveal downregulation of a large set of cell adhesion molecules and show that disruption of cadherin signaling partially phenocopies the loss of Hox5. Finally, the authors record from the motor neurons and demonstrate altered patterns of firing that they suggest arises from reduced inhibitory input.

The reviewers felt the work was especially significant for its ability to link development with the function of a physiologically important neural circuit. They had only two significant concerns they felt needed to be dealt with in the revision.

The first of these are some methodological details and controls regarding the cadherin experiments that need to be added.

The second concern regards tightening the links between the cellular mechanisms revealed (dendritic patterning, cadherin expression) and the circuit physiology shown. For example all three reviewers raised concerns about the catenin knockout experiments and the degree to which they actually reveal mechanisms of the Hox5 defects. That being said, all three reviewers also felt that the more complex genetic experiments that would directly address this concern (some of which are outlined below) might well be too involved to fit within the scope of the current manuscript. We agreed these experiments were not required to support the major points of the paper even though they would further strengthen the argument. As a result, we decided to spell out these concerns to you and leave it to you to decide what way (e.g. with experiments, with support from the literature) you see best to respond.

Essential revisions:

Methodological details (required):

In Figure 4, the authors demonstrated a unique combination of Type 2 Cadherins (Cdhs) enriched in these MNs. The significant validation was primarily carried out using in situ hybridization. There are three issues that I would like to see improvements. First, the entire set of Cdh in situ experiments showed very high background and low signals – these might be the accurate representations of the weak signals or some choice of anti-sense sequences against each and different Cdhs. The authors should include all anti-sense sequences that they chose for the study, especially this figure. I couldn't locate such information from the current PDF unless they were deposited at other locations. Generally, I would recommend sequences different from Allen Institute's short sequences (~400bp) for Cdhs – full length, or at least the complete ECC sequences should be covered. Second, for selected and enriched candidates, the authors may consider double-color fluorescent in situ or in situ in conjunction with immunohistochemistry to show the full overlapping and quantify the coverage. Third, some of the candidates identified here, including Cdh9/10 Negr1, etc. was also independently reported by Machado et al., 2014 from wildtype embryonic tissues, while this current manuscript offered much more insights genetically and functionally, leading to future function explorations of the downstream candidates. The authors should cite this manuscript and discuss their findings in the general context.

Linking cellular mechanism to physiological circuit function (optional):

1) The data in the Bax KO mice provide an important source of evidence that it is changes in the wiring and/or function of the phrenic MNs that is key for the physiology and morphological phenotypes and not just a decrease in the number of phrenic MNs. However the authors did not show the breathing phenotypes in the Bax mice, which would have been the most useful for tying the cellular and circuit studies together.

2) Although the catenin knockout phenotypes are also interesting in the context of studying phrenic MN development, I was unsure exactly how well these data supported the Hox findings. The text states that the catenin MN knockouts die prenatally, which suggests they have more severe functional disruption than the Hox mice, but the phrenic MN phenotypes shown look less severe. This led me to question exactly how the cellular data align with the breathing phenotypes.

3) By necessity, the authors analyze the respiratory phenotypes using a hypomorphic genetic background (Hoxa5^MNΔ^; c5^het^), but most of the cellular phenotypes and transcriptome comparisons are done using the complete null (Hox5^MNΔ^). It would strengthen the paper if they could better connect the dots between these two genetic states, e.g. analyze cadherin expression in the hypomorph or carry out genetic interaction experiments to show that some of the phenotypes seen in Hoxa5^MNΔ^; c5^het^ mice get significantly worse in a catenin mutant background (het or MNΔ).

4) Along the same lines, to circumvent the complicated cell adhesion code the authors depend on double catenin mutant MNs to connect their cadherin expression observations to phrenic MN defects. But, as they point out in the Discussion, the catenin mutants presumably kill the function of all cadherins. It is both not surprising that a mutant phenotype is observed in the catenin mutants, and also doesn't help to link the specific cadherins found in their transcriptome analysis to the phenotype. Perhaps a genetic interaction experiment, or knockdown of some cadherins, would address this question.

---

## [Author Response]

Essential revisions:Methodological details (required):In Figure 4, the authors demonstrated a unique combination of Type 2 Cadherins (Cdhs) enriched in these MNs. The significant validation was primarily carried out using in situ hybridization. There are three issues that I would like to see improvements. First, the entire set of Cdh in situ experiments showed very high background and low signals – these might be the accurate representations of the weak signals or some choice of anti-sense sequences against each and different Cdhs. The authors should include all anti-sense sequences that they chose for the study, especially this figure. I couldn't locate such information from the current PDF unless they were deposited at other locations. Generally, I would recommend sequences different from Allen Institute's short sequences (~400bp) for Cdhs – full length, or at least the complete ECC sequences should be covered.

We have included a table in the revised manuscript indicating the sequences of the primers used to amplify cDNA for making antisense probes, the length of the probe and the percentage of the probe sequence located in the extracellular domain (Supplementary file 1). The majority of our probes are 800-1000bp in length. For cadherin probes we chose sequences that include a major part of the extracellular domain. We have also replaced the initial cadherin in situ images shown (now in Figure 4—figure supplement 1) with fluorescent in situ images that show reduced background and enhanced signal (see revised Figure 4).

Second, for selected and enriched candidates, the authors may consider double-color fluorescent in situ or in situ in conjunction with immunohistochemistry to show the full overlapping and quantify the coverage.

We have now performed in situ in conjunction with immunohistochemistry, using a Scip antibody to label phrenic motor neurons, for all phrenic-expressed cadherins and quantified the percentage of phrenic motor neurons that express each cadherin (see revised Figure 4 and Figure 4—figure supplement 1). We find that all PMC cadherins are expressed in over 90% of phrenic MNs.

Third, some of the candidates identified here, including Cdh9/10 Negr1, etc. was also independently reported by Machado et al., 2014 from wildtype embryonic tissues, while this current manuscript offered much more insights genetically and functionally, leading to future function explorations of the downstream candidates. The authors should cite this manuscript and discuss their findings in the general context.

Following the reviewers’ suggestion, we have included a citation of Machado et al. in the revised Results and discuss their findings in the revised Discussion.

Linking cellular mechanism to physiological circuit function (optional):1) The data in the Bax KO mice provide an important source of evidence that it is changes in the wiring and/or function of the phrenic MNs that is key for the physiology and morphological phenotypes and not just a decrease in the number of phrenic MNs. However the authors did not show the breathing phenotypes in the Bax mice, which would have been the most useful for tying the cellular and circuit studies together.

Per the reviewers’ suggestion, we attempted to perform plethysmography experiments in *Hoxa5^MNΔ^; c5^het^; Bax-/- mice*, but unfortunately we were unable to recover *Hoxa5^MNΔ^; c5^het^; Bax-/-* mice at postnatal stages. Due to our breeding scheme (Bax KO mice have to be bred as hets because male KO are sterile and the females are unhealthy) and the ~50% perinatal lethality we observe for these mice (similar to *Hoxa5^MNΔ^; c5^het^*mice), the probability of generating viable mice of the correct genotype is very low (1/32), making this experiment impractical. Since *Bax* deletion does not rescue perinatal lethality, diaphragm innervation (Philippidou et al.,2012), PMC organization, gene expression or MN output (this manuscript), we think it is unlikely that it would rescue breathing behaviors.

2) Although the catenin knockout phenotypes are also interesting in the context of studying phrenic MN development, I was unsure exactly how well these data supported the Hox findings. The text states that the catenin MN knockouts die prenatally, which suggests they have more severe functional disruption than the Hox mice, but the phrenic MN phenotypes shown look less severe. This led me to question exactly how the cellular data align with the breathing phenotypes.

We find that mice with either complete loss of Hox5 function (*Hox5^MNΔ^*mice) or catenin function (*βγ-cat^MNΔ^*mice) do not survive past birth, suggesting that disruptions in either pathway lead to severe breathing phenotypes and have similar functional consequences. We also see similar disruptions in cell body organization (compare Figure 2—figure supplement 2A-D to Figure 5B-C) and dendritic orientation (compare Figure 2C-E to Figure 5D-E) in *Hoxa5^MNΔ^; c5^het^* and *βγ-cat^MNΔ^*mice, indicating that *βγ-cat^MNΔ^*mice are also likely to exhibit changes in phrenic MN output. Unfortunately, the early embryonic lethality of *βγ-cat^MNΔ^*mice precludes behavioral and electrophysiological analysis and we are currently developing additional mouse models (single and compound cadherin mutants) to further assess the role of cadherin signaling in phrenic MN connectivity and output. The fact that *βγ-cat^MNΔ^*mice show normal diaphragm innervation suggests that the functional disruption and perinatal lethality in *βγ-cat^MNΔ^*mice are a result of disrupted connectivity at the level of the cell bodies/dendrites and not at the axon/NMJ. Because Hox5 mutations affect both cell body/dendritic topography and diaphragm innervation, it is likely that both contribute to perinatal lethality and breathing phenotypes.

3) By necessity, the authors analyze the respiratory phenotypes using a hypomorphic genetic background (Hoxa5^MNΔ^; c5^het^), but most of the cellular phenotypes and transcriptome comparisons are done using the complete null (Hox5^MNΔ^). It would strengthen the paper if they could better connect the dots between these two genetic states, e.g. analyze cadherin expression in the hypomorph or carry out genetic interaction experiments to show that some of the phenotypes seen in Hoxa5^MNΔ^; c5^het^ mice get significantly worse in a catenin mutant background (het or MNΔ).

Following the reviewers’ suggestion, we have now included an analysis of CAM expression in hypomorphic (both *Hoxa5^MNΔ^; c5^het^* and *Hoxa5^MNΔ^; c5^het^; Bax-/-*) mice (see Figure 4—figure supplement 1A). Similar to *Hox5^MNΔ^; Bax-/-* mice, we see a significant downregulation of all PMC-specific CAMs in the hypomorphic genetic backgrounds. There is some residual expression of certain CAMs (e.g *cdh9, Lsamp*), which could account for the less severe phenotype of *Hoxa5^MNΔ^; c5^het^*mice when compared to *Hox5^MNΔ^*mice.

4) Along the same lines, to circumvent the complicated cell adhesion code the authors depend on double catenin mutant MNs to connect their cadherin expression observations to phrenic MN defects. But, as they point out in the Discussion, the catenin mutants presumably kill the function of all cadherins. It is both not surprising that a mutant phenotype is observed in the catenin mutants, and also doesn't help to link the specific cadherins found in their transcriptome analysis to the phenotype. Perhaps a genetic interaction experiment, or knockdown of some cadherins, would address this question.

Our analysis of the catenin mutants establishes for the first time that cadherin signaling is required for phrenic MN organization and dendritic topography. This is an important and novel finding as, despite the importance of these two key phrenic MN properties, the molecular pathways that underlie PMC organization were unknown. Furthermore, while PMC neurons specifically express a number of other CAMs (as shown in Figure 4), we have established that cadherins are playing a predominant role in PMC clustering. The implication of cadherins in PMC development shown in this manuscript is an important first step in further dissecting these molecular pathways. We are currently generating and analyzing single and compound cadherin mutants to determine the contribution of distinct cadherin family members to phrenic MN organization and we will continue to pursue this direction for future studies. However, given the significant time investment required for these experiments, we believe they are outside the scope of the current study.